# Software sustainability of global impact models

Emmanuel Nyenah[1], Petra Döll[1, 2], Daniel S. Katz[3], and Robert Reinecke[4]

[1]Institute of Physical Geography, Goethe-University Frankfurt, 60438 Frankfurt am Main, Germany
[2]Senckenberg Biodiversity and Climate Research Centre (SBiK-F), 60438 Frankfurt am Main, Germany
[3]NCSA & CS & ECE & iSchool, University of Illinois Urbana-Champaign, Urbana, IL, 61801, USA
[4]Institute of Geography, Johannes Gutenberg-University Mainz, 55128 Mainz, Germany

*Correspondence to*: Emmanuel Nyenah (Nyenah@em.uni-frankfurt.de)

**Abstract.** Research software for simulating Earth processes enables estimating past, current, and future world states and guides policy. However, this modelling software is often developed by scientists with limited training, time, and funding, leading to
software that is hard to understand, (re)use, modify, and maintain, and is, in this sense, non-sustainable. Here we evaluate the sustainability of global-scale impact models across ten research fields. We use nine sustainability indicators for our assessment. Five of these indicators – documentation, version control, open-source license, provision of software in containers, and the number of active developers – are related to best practices in software engineering and characterize overall software sustainability. The remaining four – comment density, modularity, automated testing, and adherence to coding standards –
contribute to code quality, an important factor in software sustainability. We found that 29% (32 out of 112) of the global impact models (GIMs) participating in the Inter-Sectoral Impact Model Intercomparison Project were accessible without contacting the developers. Regarding best practices in software engineering, 75% of the 32 GIMs have some kind of documentation, 81% use version control, and 69% have open-source license. Only 16% provide the software in containerized form which can potentially limit result reproducibility. Four models had no active development after 2020. Regarding code
quality, we found that models suffer from low code quality, which impedes model improvement, maintenance, reusability, and reliability. Key issues include a non-optimal comment density in 75%, insufficient modularity in 88%, and the absence of a testing suite in 72% of the GIMs. Furthermore, only 5 out of 10 models for which the source code, either in part or in its entirety, is written in Python show good compliance with PEP 8 coding standards, with the rest showing low compliance. To improve the sustainability of GIM and other research software, we recommend best practices for sustainable software
development to the scientific community. As an example of implementing these best practices, we show how reprogramming a legacy model using best practices has improved software sustainability.

## 1 Introduction

Simulation models of the Earth system are essential tools for scientists and their outcomes are relevant for decision-makers (Prinn, 2013). They improve our understanding of complex subsystems of the Earth (Prinn, 2013; Warszawski et al., 2014) and enable us to perform numerical experiments that would otherwise be impossible in the real world, e.g., exploring future pathways (Kemp et al., 2022; Satoh et al., 2022; Wan et al., 2022). A specific class of simulation models of the Earth called impact models enables us to quantitatively estimate the potential impacts of climate change on, e.g., floods (Sauer et al., 2021),

droughts (Satoh et al., 2022), and food security (Schmidhuber and Tubiello, 2007). These impact models also quantify the historical development and current situation of key environmental issues such as water stress, wildfire hazard, and fish population. The outputs of these models whether data, publications or reports thus provide crucial information for policymakers, scientists, and citizens. The central role of impact models can be seen in model intercomparison efforts of ISIMIP (Inter-Sectoral Impact Model Intercomparison Project) (ISIMIP, 2024; Warszawski et al., 2014) which encompasses

more than 130 sectoral models (Frieler and Vega, 2019). ISIMIP uses bias-corrected climate forcings to assess the potential impacts of climate change in controlled experiments, and their outputs provide valuable contributions to the Intergovernmental Panel on Climate Change reports (Warszawski et al., 2014).

      Impact models quantify physical processes related to specific components of the Earth system at various spatial and temporal

scales by using mathematical equations. The complexity of impact models is influenced by the complexity of the included physical processes, the choice of the perceptual and mathematical model, the computational effort needed for simulation, as well as their spatial-temporal resolution and spatial extent of the simulated domain (Azmi et al., 2021; Wagener et al., 2021). This complexity can result in models with very large source codes (Alexander and Easterbrook, 2015).

The software for these impact models is categorized as research software, which includes "source code files, algorithms, computational workflows, and executables developed during the research process or for a research objective" (Barker et al., 2022). Impact modelling research software is predominantly developed and maintained by scientists without formal training in software engineering (Barton et al., 2022; Carver et al., 2022; Hannay et al., 2009; Reinecke et al., 2022). Most of these researchers are self-taught software developers (Nangia and Katz, 2017; Reinecke et al., 2022) with little knowledge of

software requirements (specifications and features of software), industry-standard software design patterns (Gamma et al., 1994), good coding practices (e.g., using descriptive variable names), version control, software documentation, automated testing and project management practice (e.g. agile) (Carver et al., 2013, 2022; Hannay et al., 2009; Reinecke et al., 2022). We hypothesize that this leads to the creation of source code that is not well-structured, not easily (re)usable, difficult to modify and maintain, has scarce internal documentation (code comments) and external documentation (e.g. manuals, guides, and

tutorials), and poorly documented workflows.

Research software that suffers from these shortcomings is likely difficult to sustain and has severe drawbacks for scientific research. For example, it can impede research progress, decrease research efficiency, and hinder scientific progress, as implementing new ideas or correcting mistakes in code that is not well-structured is more difficult and time-consuming. In addition, it increases the likelihood of erroneous results, thereby reducing reliability and hindering reproducibility (Reinecke et al., 2022). We argue that these harmful properties can be averted, to some extent, with sustainable research software.

There are various interpretations of the meaning of "sustainable research software". Anzt et al. (2021) define research software as software that is maintainable, extensible, flexible (adapts to user requirements), has a defined software architecture, is testable, has comprehensive in-code and external documentation, and is accessible (the software is licensed as Open Source with a digital object identifier (DOI) for proper attribution) (Anzt et al., 2021). For example, NumPy (https://numpy.org/) is a widely used scientific software package that exemplifies many of these qualities (Harris et al., 2020). Although NumPy is not an impact model, it is an exemplar of sustainable research software; it is open-source, maintains rigorous version control and testing practices, and is extensively documented, making it highly reusable and extensible for the scientific community.

Katz views research software sustainability as the process of developing and maintaining software that continues to meet its purpose over time (Katz, 2022). This includes adding new capabilities as needed by its users, responding to bugs and other problems that are discovered, and porting to work with new versions of the underlying layers, including software as well as new hardware (Katz, 2022). Both definitions share common aspects like the adaptation to user requirements but differ in scope and perspective. Katz's definition is more user-oriented, focusing on the software's ability to continue meeting its purpose over time. On the other hand, Anzt et al.'s definition is more developer-oriented, aiming to improve the quality and robustness of research software. We chose to adopt Anzt et al.'s definition in the following because it provides measurable qualities relevant to this study. In contrast, Katz's definition is more challenging to measure and evaluate but is likely closer to the reality of software development. For example, one of the models in our analysis is more than 25 years old (Nyenah et al., 2023) and thus certainly was sustained during that period, while at the same time, it does not meet some sustainability requirements of Anzt et al.'s definition. It is possible that such software can be sustained but requires substantial additional resources.

Recent advances in developing sustainable research software have led to a set of community standard principles: FAIR (findable, accessible, interoperable, reusable) for research software (FAIR4RS), aimed towards increasing transparency, reproducibility, and reusability of research (Barker et al., 2022; Chue Hong et al., 2022). Software quality which impacts sustainability overlaps with the FAIR4RS principles, particularly reusability, but is not directly addressed by them (Chue Hong et al., 2022). Reusable software here means software can be understood, modified, built upon, or incorporated into other software (Chue Hong et al., 2022). A high degree of reusability is therefore important for efficient further development and improvement of research software, and thus for scientific progress. However, many models are not FAIR (Barton et al., 2022). To our knowledge, research software sustainability in Earth System Sciences has not been evaluated before.

As an example of complex research software in the Earth System Sciences, in this study, we assess the sustainability of the software of global impact models (GIMs) that participate in the ISIMIP project to investigate factors that contribute to sustainable software development. The GIMs belong to the ten research fields (or impact sectors): agriculture, biomes, fire, fisheries, health, lakes, water (resources), water quality, groundwater, and terrestrial biodiversity. In our assessment, we consider nine indicators of research software sustainability, five of them related to best practices in software engineering and four related to source code quality. We further provide first-order cost estimates required to develop these GIMs but do not address the cost of re-implementing or making code reproducible versus the cost of maintaining old code in this study. We also demonstrate how reprogramming legacy software using best practices can lead to significant improvements in code quality and thus sustainability. Finally, we offer actionable recommendations for developing sustainable research software for the scientific community.

## 2 Methods

### 2.1 Accessing GIM Source code

ISIMIP manages a comprehensive database of participating impact models (available in an Excel file at https://www.isimip.org/impactmodels/download/), which provides essential information such as model ownership, name, source code links, and simulation rounds. Initially, we identified 375 models across five simulation rounds (fast track, 2a, 2b, 3a, and 3b). As the focus of our analysis is on global impact models, we sorted the models by spatial domain and filtered out models operating at local and regional scales, resulting in a subset of 264 GIMs. We then removed duplicate models, prioritizing the most recent versions for inclusion, resulting in 112 unique models. For models with available source links, we obtained their source code directly. In instances where source links were not readily available, we conducted manual searches for source code by referring to code availability sections in reference papers. Additionally, we searched for source code using model names along with keywords such as "GitHub" and "GitLab" using the Google search engine. As of April 2024, 32 out of the 112 unique model source codes were accessible either through direct links from the ISIMIP database or via manual searches on platforms like GitHub and GitLab, as well as in code availability sections of reference papers. However, it's important to note that our sample may suffer from a "survivor bias," as we are not investigating models that are no longer in use (GIMs that couldn't be sustained over time). This bias could potentially skew our analysis towards models that have survived i.e., they are still in use and their source code is accessible. Due to time constraints, we refrained from contacting developers for models that were not immediately accessible.

### 2.2 Research software sustainability indicators

We examine nine indicators of research software sustainability, distinguishing five indicators related to the best practice in software engineering and four indicators of source code quality (Table 1).

**Table 1: Indicators used for the assessment of research software sustainability**

| No. | Indicator | Description |
| --- | --- | --- |
| *Best practices in software engineering* | | |
| 1 | Documentation | Enables software use and also makes software maintenance easier (Wilson et al., 2014). |
| 2 | Version control | Provides transparency and traceability throughout the software development lifecycle and enables collaboration between developers as well as user communities (Wilson et al., 2014). |
| 3 | Use of an open-source license | Allows code copying and reuse. This openness fosters a collaborative environment where the user community can provide valuable feedback and support. Users can potentially contribute to the software's development and maintenance, enhancing its overall quality (Jiménez et al., 2017). |
| 4 | Number of active developers | Prevent single points of failure in the development process and make software development as well as maintenance easier (Long, 2006). |
| 5 | Containerization | Makes the software easy to install and facilitates reproducibility (Nüst et al., 2020; Wilson et al., 2014). |

*Source code quality*

| | | |
|---|---|---|
| 6 | Public availability of an (automated) testing suite | Shows that software functionality can be or was tested. |
| 7 | Compliance with coding standards (e.g. PEP 8) | Improves code quality, readability and makes maintenance easier (Capiluppi et al., 2009; Simmons et al., 2020; Wang et al., 2008). |
| 8 | Comment density | Precursor to software maintainability and re-usability (Arafat and Riehle, 2009; He, 2019; Stamelos et al., 2002). |
| 9 | Modularity | Necessary for extensible and flexible research software (Sarkar et al., 2008; Stamelos et al., 2002). |


In the following, we describe the indicators and their rationale and how we evaluated the GIMs with respect to each indicator.


*Documentation.* Documentation is crucial for understanding and effectively utilizing software (Wilson et al., 2014). This includes various materials such as manuals, guides, tutorials that explain the usage and functionality of the software as well reference model description papers. When assessing documentation availability, relying solely on a reference model description paper may be insufficient, as it may not provide the level of detail necessary for the effective utilization and

maintenance of the research software. All GIMs used in this assessment have an associated description or reference paper (see supplementary file ISIMIP_models.xlsx). Therefore, in addition to the reference model paper we checked for available manuals, guides, readme files, and tutorials. We consider any of these resources, alongside the reference model paper, as documentation for the model. These resources provide essential information such as user, contributor, and troubleshooting guides, which are valuable for model usage and maintenance. In our assessment, we searched within the source code and

official websites (if available). We also utilized the Google search engine to find model documentation by inputting model names along with keywords such as 'documentation,' 'manuals,' 'readme,' 'guides,' and 'tutorials'.

*Version control.* Version control systems such as Git and Mercurial facilitate track changes, and collaborative development, and provide a history of software evolution. To assess whether GIMs use version control for development, we focused on

commonly used open-source version control hosting repositories such as GitLab, GitHub, BitBucket, Google Code, and Source Forge. The hostname such as "github" or "gitlab" in the source link of models provides clear indications of version control adoption in their development process. For other models, we searched within the Google search engine using model names and keywords such as "Bitbucket", "Google Code", and "Source Forge". While we focus on identifying the use of version control systems, evaluating how version control was implemented during the development process — such as the use of

modular commits, pull requests, discussions, and proper versioning — is a finer analysis that falls beyond the scope of this study. However, such practices are crucial for ensuring high-quality software development and collaborative practices.

*Use of an open-source license*. We determined the existence of open-source licenses by checking license files within

repositories or official websites against licenses approved by the Open Source Initiative (OSI) (https://opensource.org/licenses). Specifically, we looked for licenses that conform to the Open Source Definition, which ensures that software can be freely used, modified, and shared (Colazo and Fang, 2009; Rashid et al., 2019). There are two major categories of open-source licenses: permissive licenses, such as MIT or Apache, that allow for minimal restrictions on how the software can be used (e.g., providing attribution), and copyleft licenses, like GPL, that require derivatives to maintain

the same licensing terms (Colazo and Fang, 2009; Rashid et al., 2019). Although these licenses differ in their terms, both contribute to collaboration and transparency. In this study, we only check if the software is open-source, regardless of the type of open-source license.

*Number of active developers*. The presence of multiple active developers serves as a safeguard against halts within the

development process. In instances where a sole developer departs or transitions roles, the absence of additional developers could lead to disruptions or challenges in maintaining and advancing the software. We measured the number of active developers by counting the individuals who made commits or contributions to the projects codebase within the period 2020-2024. A higher number of developers indicates a greater capacity for bug review (enhancing source code quality) and code maintenance. It can also lead to more frequent updates to the source code. On the other hand, the absence of active developers

suggests potential stagnation in software evolution, possibly impacting the relevance and usability of the software.

*Containerization*. Containerization provides convenient ways to package and distribute software, facilitating reproducibility and deployment. It encapsulates an application along with its environment, ensuring consistent operation across various platforms (Nüst et al., 2020). Despite these benefits, containerization in high-performance computing systems encounters

challenges like performance, prompting the proposal of solutions (Zhou et al., 2023). Some popular containerization solutions include Docker (https://www.docker.com/) and Apptainer (https://apptainer.org/). There are also cloud-supported container solutions such as Binder (https://mybinder.org/) with the capacity to execute a model with the computational environment requirements analogous to the concept of analysis-ready data and cloud-optimized formats for datasets (Abernathey et al.,

2021). To evaluate the availability of container solutions, we conducted searches through reference papers, official websites,
and software documentation for links to container images or image-building files such as "Dockerfiles", and "Apptainer definition file (.def file)". In addition, we also searched through source code repositories to identify the previous stated images or image-building files. Lastly, we utilize the Google search engine, inputting the name of the GIM, the sector, and keywords such as "containerization", to ascertain if any other containerized solutions exist.

*Public availability of an (automated) testing suite.* Test coverage, which verifies the software's functionality, is the property of actual interest. However, research software may have an automatic testing suite but not provide information on test coverage or test results. As a practical approach, we consider the availability of a testing suite as a proxy for the ability to test software functionality. By examining testing suites within repositories, we gain insights into the developers' commitment to software testing, which contributes to enhancing software quality.


*Compliance with coding standards.* Coding standards are a set of industry-recognized best practices that provide guidelines for developing software code (Wang et al., 2008). Analysing the conformance to these standards can be complex, particularly when the source code is written in multiple languages. Different languages may have various coding styles or style guides. For instance, multiple style guides are available and accepted by the Julia community (JuliaReachDevDocs, 2024). As an example
analysis, we focused on GIMs containing Python in their source code as it is one of the most prevalent languages used in development. The tool used, known as Pylint , is designed to analyze Python code for potential errors and adherence to coding standards (Molnar et al., 2020; Obermüller et al., 2021). Pylint evaluates source files for their compliance with PEP8 conventions. To quantify adherence to this coding standard, it assigns a maximum score of 10 as perfect compliance but has no lower bound (Molnar et al., 2020). We consider scores below 6 as indicative of weak compliance as code contain several
violations.

*Comment density.* Good commenting practice is valuable for code comprehension and debugging. Comment density is an indicator of maintainable software (Arafat and Riehle, 2009; He, 2019). Comment density is defined as

$$Comment\ density = \frac{Number\ of\ lines\ of\ comment}{Total\ lines\ of\ code} \tag{1}$$

Here, the total lines of code (TLOC) include both comments and source lines of code (SLOC) (SLOCCount, 2024). SLOC is defined as the physical non-blank, non-comment line in a source file. Arafat et al. (2009) and He (2019) suggest that comment density between 30-60% may be optimal. For most programming languages, this range is considered to represent a compromise between providing sufficient comments for code explanation and having too many comments that may distract from the code logic (Arafat and Riehle, 2009; He, 2019).


*Modularity.*  Researchers typically pursue new knowledge by asking and then attempting to answer new research questions. When the questions can be answered via computation, this requires either building new software, adding new source code, or modifying existing source code. Addition and modification of source code are more easily achieved if the software has a modular structure that is implemented as extensible and flexible software (McConnell, 2004). Therefore, modularity is chosen as another indicator for research software sustainability. Modular programming is an approach where source codes are organised into smaller and well-manageable units (modules) that execute one aspect of the software functionality, such as the computation of evapotranspiration in a hydrological model (Sarkar et al., 2008; Trisovic et al., 2022). The aim is that each module can be easily understood, modified, and reused. Depending on the programming language, a module can be a single file (e.g. Python) or a set of files (e.g. C++).

To assess the modularity of research software, we use the TLOC per file as a metric. This metric reflects the organization of the source code into modules, each performing a specific function (Sarkar et al., 2008; Trisovic et al., 2022). We opted for this approach over measuring TLOC per function or subroutine due to variations in programming languages and the challenges associated with accurately measuring different functions using program-specific tools. For instance, in Python, a module that contains significantly more TLOC than usual (here over 1,000 TLOC) likely includes multiple functions. These functions may perform more than one aspect of the software's functionality, such as reading input files and computing other functions (e.g. evapotranspiration function), which contradicts the principle of modularity. Keeping the length of code in each file concise also enhances readability.

The ideal number of TLOC per file can vary with the language, paradigm (e.g., procedural or object-oriented), and coding style used in a software project (Fowler, 2019; McConnell, 2004). However, a common heuristic is to keep the code size per file under 1,000 lines to prevent potential performance issues such as crashes or slow program execution with some integrated development environments (IDEs) (Fowler, 2019; McConnell, 2004). IDEs are software applications that provide tools like code editors, debuggers, and build automation tools. As reported by Trisovic et al. (2022), based on interviews with top software engineers, a module with a single file should contain at least 10 lines of code, consisting of either functions or statements (Trisovic et al., 2022). We used this heuristic as a criterion for good modularity, assuming that 10-1,000 TLOC per file indicates adequate modularity. We also varied the upper bounds of the total lines of code to 5,000 and 500 to investigate how modularity changes across models and sectors.

### 2.3 Source code counter

To count SLOC, comment lines, and TLOC of computational models, the counting tool developed by Ben Boyter (https://github.com/boyter50/scc) was used (Boyter, 2024). This tool builds on the industrial standard source code counter tool called SLOCCount (Source Lines of Code Count) (SLOCCount, 2024).

## 2.4 Software cost estimation

The cost of developing research software is mostly unknown and depends on many factors, such as project size, computing infrastructure, and developer experience (Boehm, 1981). A model that attempts to estimate the cost of software development is the widely used Constructive Cost Model (COCOMO) (Boehm, 1981; Sachan et al., 2016), which computes the cost of commercial software by deriving the person-months required for developing the code based on the lines of code. Sachan et al. (2016) used the TLOC and effort estimates of 18 very large NASA projects (Average TLOC = 35,000) to optimise the parameters of the COCOMO regression model (Sachan et al., 2016). Effort in person months is estimated following Eq. (2):

$$Effort = 2.022817(kTLOC)^{0.897183} \tag{2}$$

where total lines of code are expressed in 1,000 TLOC (kTLOC) (Sachan et al., 2016). We use this cost model to estimate the cost of GIMs.

## 3 Results and Discussion

### 3.1 GIM programming languages and access points

The source code of the 32 GIMs is written in 10 programming languages (Fig. 1a). Fortran and Python are the most widely used, with 11 and 10 models, respectively. The dominance of Fortran stems from its performance, and the fact that it is one of the oldest programming languages designed for scientific computing (Van Snyder, 2007), and was the main such language used at the time some of the GIMs were originally built. This specialization makes it particularly suitable for tasks involving numerical simulations and complex computations. On the other hand, Python enjoys popularity among model developers due to readability, large user community, and rich ecosystem of packages, including those supporting parallel computing. R, C++ and C follow with 5, 5, and 4 models respectively (Fig. 1a). GIMs may employ one or more programming languages to target specific benefits the programming languages offer, such as readability and performance. For example, one of the studied models, HydroPy, written in Python, enhances its runtime performance by integrating a routing scheme built in Fortran (Stacke and Hagemann, 2021).

We find that 24 (75%) of the readily accessible 32 GIMs were hosted on GitHub (Fig. 1b). The rest are made available on GitLab (2, or 6%), Zenodo (4, or 12%), or the official website of the model (2, or 6%) (Fig. 1b).
We note that for one of the GIMs used for analysis, WaterGAP2.2e, only part of the complete model (the global hydrology model) was accessed (Müller Schmied et al., 2021). This might be the case for other models as well.

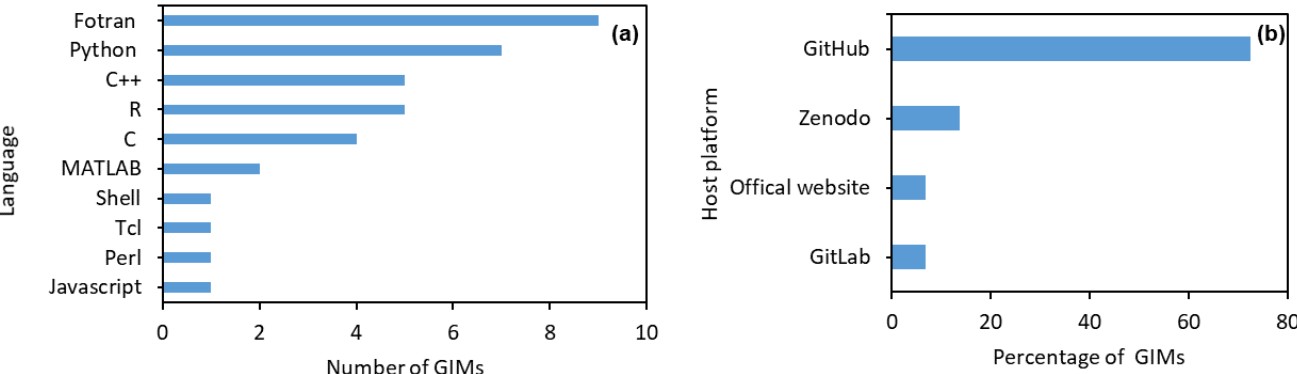


**Figure 1**: Programming languages for model development and model accessibility. (a) Bar plots showing programming languages used for developing 32 global impact models. (b) Bar plot showing open-source hosting platforms where 32 global impact models were accessed

### 3.2 Indicators of Software Sustainability

#### 3.2.1 Software Engineering Practices

*Documentation:*

Our analysis reveals that 75% of the GIMs (24 out of 32) have publicly accessible documentation (Table 2). We observed a range of documentation formats across these GIMs. Specifically, 6 GIMs provided readme files, 13 had dedicated webpages for documentation, and 5 included comprehensive manuals (see supplementary file ISIMIP_models.xlsx). While README files tend to be more minimal and sometimes difficult to navigate, we observed that they generally contain essential information such as instructions on how to run the research software. The prevalence of documentation practices among most models underscores the importance of documenting research software. However, a notable portion (25%) of the studied models either lack documentation or documentation has not been made publicly available (Table 2).



**Table 2:** Availability of Documentation, Version Control, Open-Source License, Test Suite, and Container for 32 Global Impact Models across 10 Sectors in Earth System Science. '+', '-', 'not valid' and 'no info' represent the availability, unavailability, not OSI-approved and absence of information, respectively.

| No. | Sector | Model | Year of Latest Version | Language | Documentation | Version control | Open Source License | Test Suite | Container |
|---|---|---|---|---|---|---|---|---|---|
| 1 | Agriculture | CGMS-WOFOST | no info | Fortran | + | + | + | - | - |
| 2 | Agriculture | DSSAT-Pythia | 2024 | Python | + | + | no info | + | + |
| 3 | Agriculture | EPIC-TAMU | 2023 | Fortran | + | no info | + | - | - |
| 4 | Agriculture | LPJmL | 2024 | C and JavaScript | + | + | + | - | - |
| 5 | Agriculture | ACEA | 2024 | Python | + | no info | not valid | - | - |
| 6 | Agriculture | LPJ-GUESS | 2021 | C++ | + | no info | + | - | - |
| 7 | Biomes | CLASSIC | 2020 | Fortran | + | + | + | + | + |
| 8 | Biomes | MC2-USFS-r87g5c1 | 2022 | C++, Fortran and C | + | + | + | - | - |
| 9 | Fire | SSiB4/TRIFFID-Fire | 2021 | Fortran | - | + | no info | - | - |
| 10 | Fisheries | BOATS | no info | MATLAB | - | + | no info | - | - |
| 11 | Fisheries | DBPM | no info | R | - | + | no info | + | - |
| 12 | Fisheries | EcoTroph | no info | R | + | + | no info | - | - |
| 13 | Fisheries | FEISTY | no info | MATLAB | - | + | no info | - | - |
| 14 | Fisheries | ZooMSS | 2020 | R and c++ | + | + | + | - | - |
| 15 | Groundwater | G³M | 2018 | C++ | + | + | + | + | - |
| 16 | Groundwater | parflow | 2024 | C, Tcl, python | + | + | + | + | + |
| 17 | Lakes | ALBM | 2024 | Fortran | + | + | + | - | - |
| 18 | Lakes | GOTM | 2024 | Fortran | + | + | + | + | - |
| 19 | Lakes | SIMSTRAT-UoG | 2024 | Fortran | + | + | + | + | + |
| 20 | Terrestrial biodiversity | BioScen15-SDM-GAM/GBM | no info | R | - | + | no info | - | - |
| 21 | Terrestrial biodiversity | BioScen1.5-MEM-GAM/GBM | no info | R | - | + | + | - | - |
| 22 | Vector-borne diseases (health) | VECTRI | no info | Fortran and python | + | + | + | - | - |
| 23 | Water | CWatM | 2023 | Python | + | + | + | + | - |
| 24 | Water | DBH | 2006 | Fortran | + | no info | not valid | - | - |
| 25 | Water | HydroPy | 2021 | Python | + | no info | + | - | - |
| 26 | Water | PCR-GLOBWB | 2023 | Python | + | + | + | - | - |
| 27 | Water | WBM | 2023 | Perl | + | + | + | - | - |
| 28 | Water | WaterGAP2.2e | 2023 | C++ | - | no info | + | - | - |
| 29 | Water | VIC | 2021 | C and Python | + | + | + | + | + |
| 30 | Water | H08 | 2024 | Fortran and Shell | + | + | + | - | - |
| 31 | Water | WAYS | no info | Python | - | + | + | - | - |

| 32 | Water quality | DynQual | | 2023 | Python | + | + | no info | - | - |
|---|---|---|---|---|---|---|---|---|---|---|
| | *Total* | | | | | 24 | 26 | 22 | 9 | 5 |

*Version control:*

We find that 81% (26 out of 32) of GIMs uses Git as their version control system reflecting the widespread acceptance of Git
across the sectors (Table 2). In the remaining cases, information about the specific version control system used for these GIMs
was unavailable.

*Use of an open source license*:

Most of the research software, 69% (22 out of 32), have open-source licenses (Table 2) with the "GNU General Public License"
being the commonly used license (56%, 18 out of 32) (Fig. 2). However, the remaining 31% (10 out of 32) either have no
information on the license even though the source code is made publicly available (8 or 25% of GIMs) or uses license which
is not OSI-approved (1 GIM each with creative commons license and user agreement) (Fig. 2). This ambiguity or absence of
licensing details can deter potential users and contributors, as it raises uncertainties about the permissions and restrictions
associated with the software.


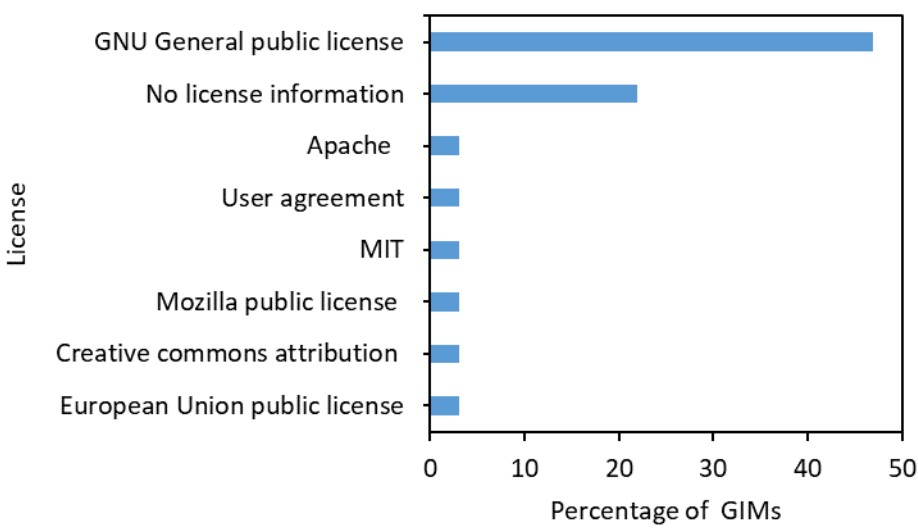

**Figure 2**: License distribution for 32 global impact models across 10 sectors. 8 (25%) GIMs lack license information, and
two (6%) GIMs have licenses that are not OSI-approved.

*Number of active developers:*

Our results reveal a diverse distribution of active developers across the GIMs. We have excluded GIMs without version control information from our results, as without could not be evaluated for this indicator, resulting in data for 26 GIMs. Notably, GIMs such as parflow, CWatM, LPJmL, and GOTM have a significant number of active developers, with 28, 12, 11, and 8 developers respectively (Fig. 3). These values correlates with the size of GIMs source code, as evidenced by TLOC (282,722 for ParFlow, 33,286 for CWatM, 136,002 for LPJmL, and 29,477 for GOTM.). However, models such as WAYS, VIC, BioScen1.5-MEM, and CGMS-WOFOST had no active developers during the considered period of 2020 to 2024 (Fig. 3).

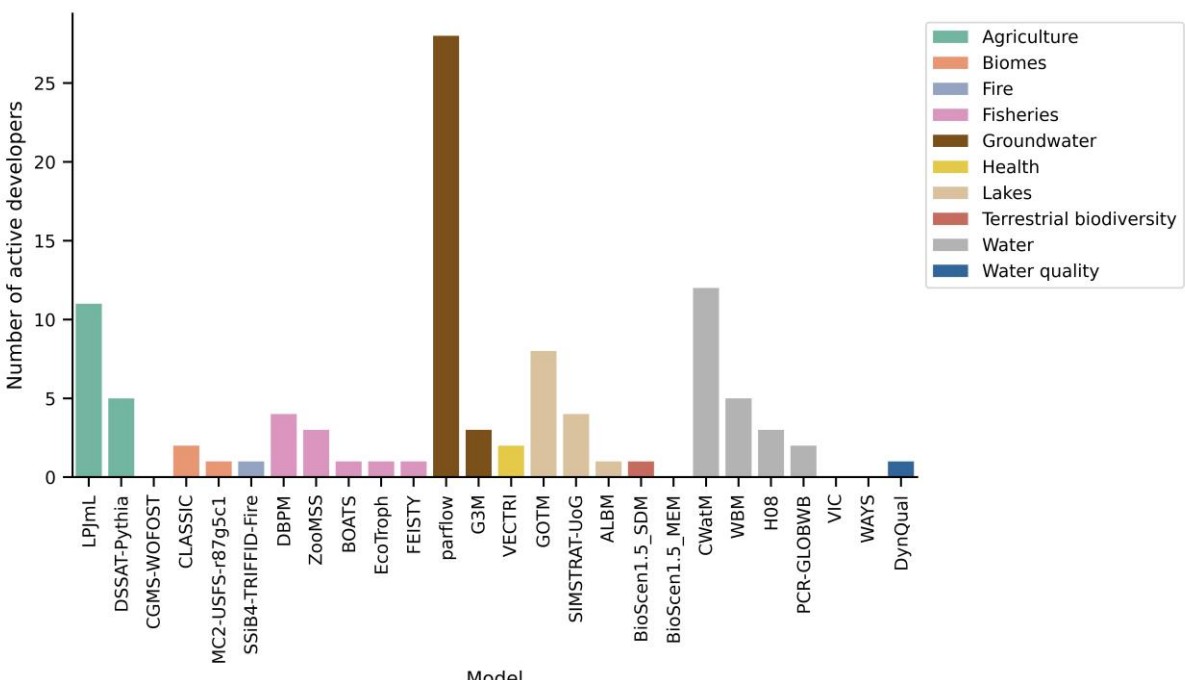

**Figure 3**: Number of active developers within 5 years (2020-2024) for 26 global impact models across 10 sectors. The results for the 6 remaining GIMs could not be measured since version control information could not be found. Zero value means no active developers within the 5 year period. The models are sorted within each sector by the number of active developers.

*Containerization:*

Only 5 (16%) of the GIMs have implemented containerized solutions (Table 2). While the CLASSIC model uses Apptainer, the other four models use Docker as their containerization technology. The CLASSIC container is shared via Zenodo, whereas the Docker containers for the remaining models are distributed through GitHub. Despite the recognized benefits of

containerization in promoting reproducible research, provisioning of the software in containers is not yet a common practice in GIM development.


### 3.2.2 Code Quality Indicators

*Public availability of an (automated) testing suite:*

Our research indicates that 28% (9 out of 32) of the examined GIMs have a testing suite in place to test the software's

functionality (Table 2). The models with test suites do not use a preferred programming language but have various languages, including Python, Fortran, R, and C++ (Table 2). While the choice of programming language can influence the ease of implementing test suites (e.g., due to the availability of testing libraries), we observe that for these complex models, which often prioritize computational performance, implementing a test suite remains essential regardless of the programming language used.  A typical test might involve ensuring that a global hydrological model such as CWatM runs without errors

with different configuration file options (e.g., different resolutions and basins) (Burek et al., 2020). However, this practice is not widespread in the development of GIMs, with the majority (72%) lacking a testing suite (Table 2). This absence of testing suites in GIM development highlights a deficiency in the developers' dedication to software testing. The presence of a testing suite could lead to more frequent testing, thereby enhancing the overall quality of the software.

*Compliance with coding standards*:

We restricted our analysis to GIMs that include Python in their source code due to challenges described in section 2.2. Among the ten models we examined, we observed varying levels of adherence to the PEP8 style guide for Python. Five models (DSSAT-Pythia, parflow, HydroPy, VIC, and WAYS) demonstrated good compliance, each achieving a lint score above 6 out of a maximum of 10 (Fig. 4). Good compliance indicates minimal PEP8 code violations. However, the remaining five models

showed lower compliance, with lint scores between 0 and 3 (Fig. 4). This suggests numerous violations leading to potential issues like poor code readability and an increased likelihood of bugs, which could hinder code maintenance.

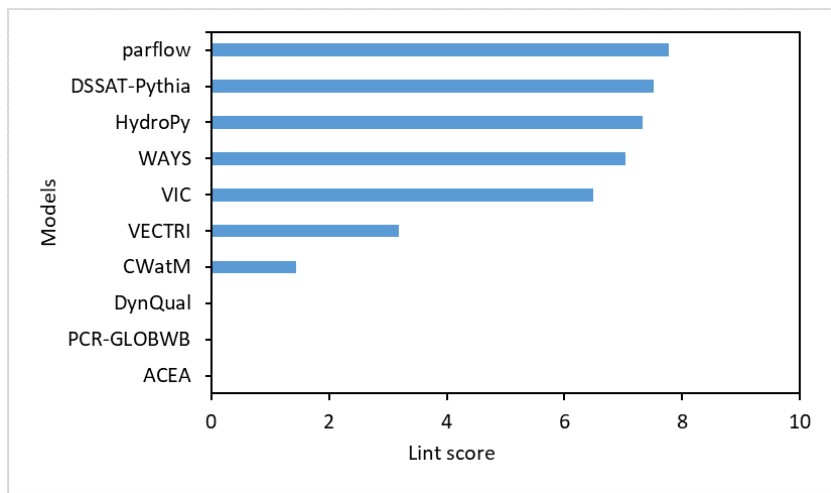

**Figure 4:** Lint scores of GIMs containing Python code.


*Comment density:*

Our results indicate that 25% (8 of 32) of the GIMs have well-commented source code, i.e. 30-60% of all source lines of code are comment lines (Fig. 5). The remaining 75% (24) of the GIMs have too few comments, which indicates that overall, commenting practice is low across the studied research fields.

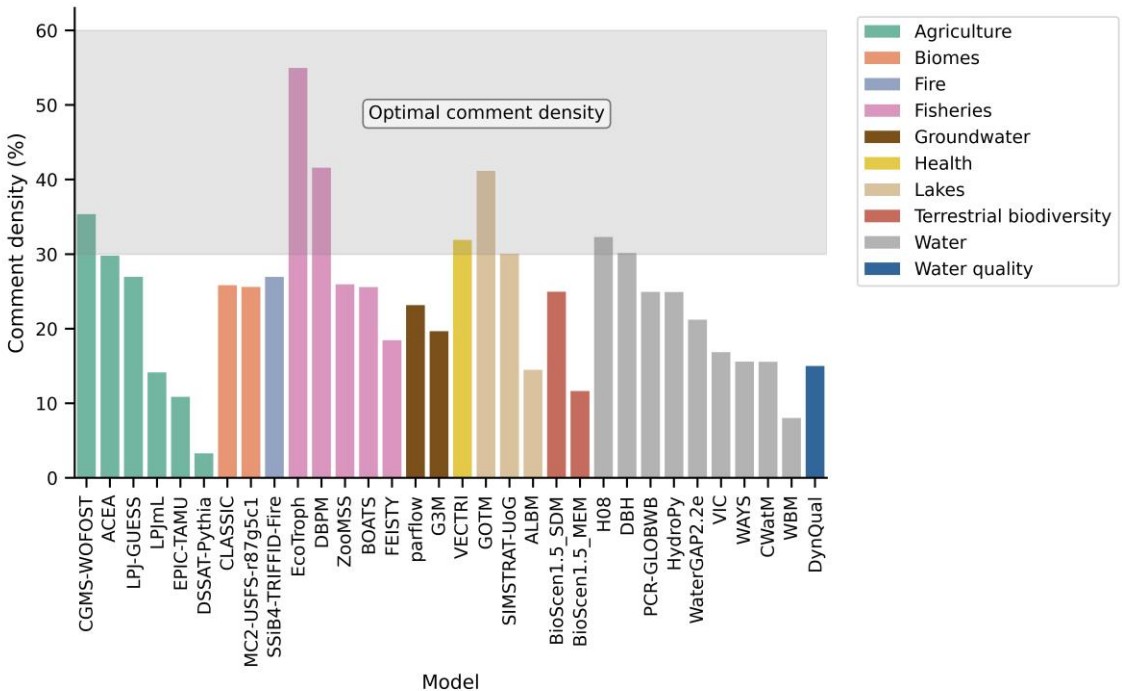

**Figure 5:** Comment density per model across 10 sectors. The grey zone denotes the optimal comment density (Arafat and Riehle, 2009; He, 2019). Models are sorted within each sector by decreasing comment density.

*Modularity:*

The investigated GIMs have TLOC values between 262 and 500,000, distributed over 6-2400 files (Fig. 6). Only 4 out of the 32 (12%) simulation models (EcoTroph, ZooMSS, HydroPy, and BioScen1.5_SDM) meet the criterion of having between 10 and 1,000 TLOC per file (Fig. 6). The remaining 28 GIMs either had at least one file exceeding 1,000 TLOC, which likely could be divided into smaller modules with distinct functionality or had at least one file less than 10 TLOC, which makes source code harder to navigate and understand, especially if the files are not well-named or documented. We also performed a sensitivity analysis by changing the criterion to 5,000 and 500 TLOC per file with the same lower limit of 10 TLOC. Nine simulation models (LPJmL, MC2-USFS-r87g5c1, EcoTroph, ZooMSS, BioScen1.5_SDM, BioScen1.5_MEM, H08, HydroPy, and DynQual) meet the 5,000-line criterion and two models (EcoTroph, ZooMSS) met the 500-line criterion (Fig. 6). Because code comments, which are included in TLOC, aid code comprehension, we also assessed modularity using the criterion of 1,000 SLOC instead of 1,000 TLOC with 10 SLOC. Three GIMs (ZooMSS, BioScen1.5_SDM, and Hydropy) meet the 10-1,000 SLOC criterion (see supplementary Fig. S1).

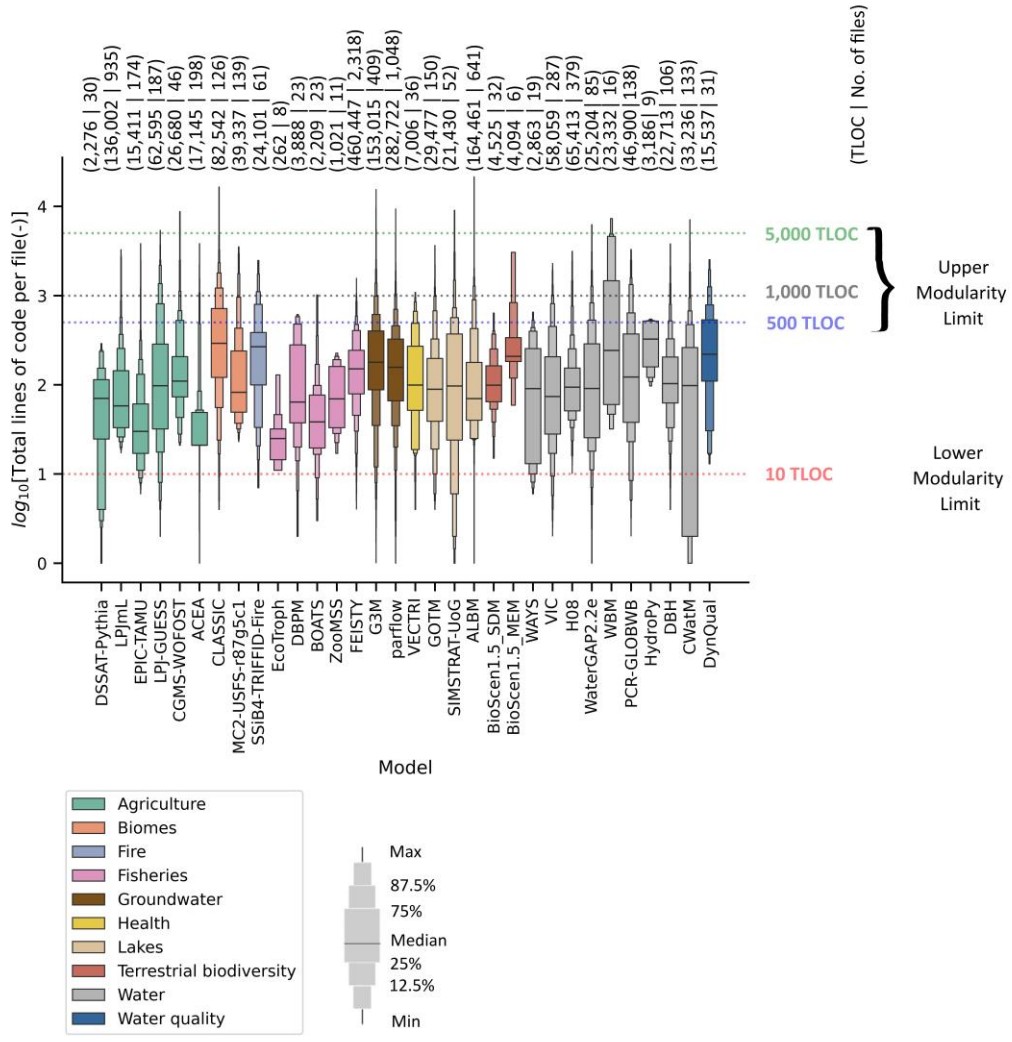

**Figure 6**: Letter value plot (Hofmann et al., 2017) of total lines of code (TLOC) per file (logarithmic scale) of 32 global impact models across 10 sectors. The dotted blue, black, and green lines show upper modularity limits, the dotted red line the lower limit. The values (x|y) in the upper section of Fig. 6, show, for each GIM, TLOC | Number of files.

### 3.3 Cost of GIM software development

To provide a rough cost estimate for the software development of the 32 impact models, we use the cost estimate model from Sachan et al. (2016) (see section 2.4) in a scenario of "*what if we would hire a commercial software company to develop the*

*source code of the global impact models*?" This cost estimate does not include developing the science (e.g., concepts, algorithms, and input data) nor costs of documenting, running, and maintaining the software, only the implementation of code. We assume that the COCOMO model is transferable to research software as the NASA projects used in cost model contain software that is similar to research software. As the TLOC of the impact model codes ranges from 262 to 500,000 TLOC (Fig. 7), the effort required to produce these models ranges from 1 to 495 person-months (Fig. 7). With a small additive change of $\pm 0.1$ of the COCOMO model coefficients, the range of estimated effort changes to 1 to 255 person-months in the case of -0.1 , and to 1 to 960 person-months in the case of +0.1 (Supplementary Fig. S2).

The results suggest that these complex research software programs are expensive tools that require adequate funding for development and maintenance to make them sustainable. This is consistent with previous studies that have highlighted funding challenges for developing and maintaining sustainable research software in various domains (Carver et al., 2013, 2022; Eeuwijk et al., 2021; Merow et al., 2023; Reinecke et al., 2022). Merow et al. (2023) also emphasized that the accuracy and reproducibility of scientific results increasingly depend on updating and maintaining software. However, the incentive structure in academia for software development — and especially maintenance — is insufficient (Merow et al., 2023).

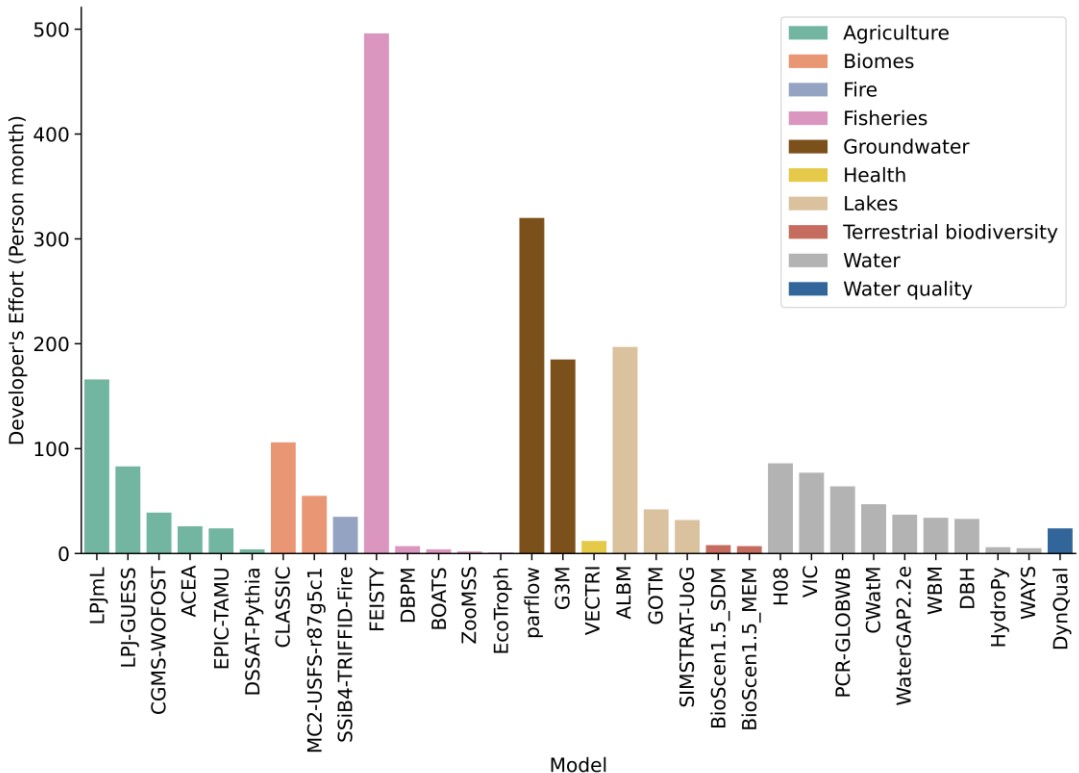

**Figure 7 :** Effort estimates of 32 global impact models across 10 sectors. Models are sorted within each sector by decreasing amount of developer's effort.

**3.4 Case Study: Reprogramming legacy simulation models with best practices**

Legacy codes often suffer from poor code readability and poor documentation, which hinder their maintenance, extension, and reuse. To overcome this problem, some of GIMs such as HydroPy (Stacke and Hagemann, 2021; Stacke, Tobias and Hagemann, Stefan, 2021) were reprogrammed, while others (e.g., WaterGAP, Nyenah et al., 2023) are in the process of being reprogrammed. We compared the legacy global hydrological model MPI-HM (in Fortran) and its reprogrammed version HydroPy (in Python) in terms of the sustainability indicators. The reprogrammed model has improved modularity (Fig. 8a),

which supports source code modification and extensibility. HydroPy has good compliance with the PEP8 coding standard, which improves readability and lower the likelihood of bugs in source code (Fig. 4). It  has an open-source license and a persistent digital object identifier, which makes it easier to cite (Editorial, 2019). This research software refers to its associated publication for information and instructions on Zenodo to setup and run Hydropy. A software testing suite and container are not yet available.

We find that HydroPy has a comment density of 25% (Fig. 8b), which is below the desired 30-60% range, but the developers argue that "the code is self-explanatory and comments are added only when necessary" (Stacke, 2023). MPI-HM has more comments (49%, Fig. 8b) because of its legacy Fortran code that limits variable names to a maximum length of 8 characters, so they have to be described in comments. Another reason is that the MPI-HM developers kept track of the file history in the header, which adds to the comment lines in MPI-HM. This raises a question: *Is the comment density threshold metric still valid*

*if a code is highly readable and comprehensive*? The need for comments can depend on the language's readability (Python vs. Fortran), the complexity of the implemented algorithms and concepts, and the coder's expertise. While a highly readable and well-structured code might require fewer explanatory comments, the definition of "readable" itself can be subjective and context-dependent. Nevertheless, comment density remains a valuable metric, especially for code written by novice developers.

The HydroPy model is a great starting point for sustainable research software development, as it illustrates the application of the sustainability indicators. Reprogramming legacy code not only allows developers to use more descriptive variable names, which increases code readability and maintainability, but also enables them to share their code and documentation with the scientific community through open source platforms and tools. This practice enhances transparency and accountability, as the code can be inspected, verified, and reproduced by others. Reprogramming legacy code with best practices always improves

code quality, which makes software more sustainable.

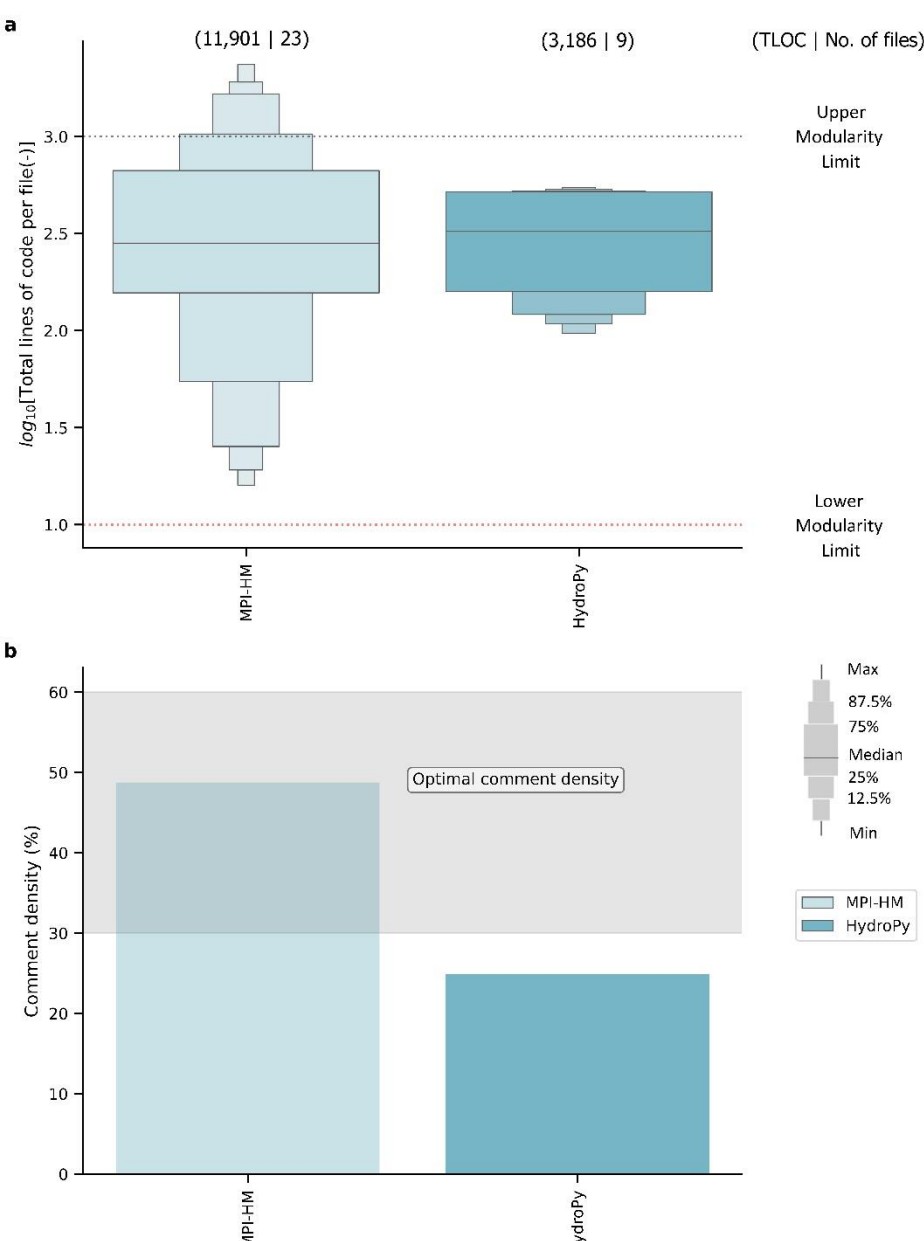

**Figure 8:** Modularity and commenting practice of a legacy (MPI-HM) and reprogrammed (HydroPy) global simulation model. (a) Letter value plot of total lines of code per file (logarithmic scale) of each model. The dotted black (red) line shows the upper (lower) modularity limit defined as the maximum of 1000 (minimum of 10) total lines of code per file. The values (x|y) shown in the upper section of Fig. 8a correspond to (TLOC | Number of files per model). (b) Comment density per model. The grey zone in Fig. 8b denotes the optimal comment density.

## 4 Limitations

Our study has limitations in the following regards. In the interest of timely analysis, we did not contact the developers of models that were not readily available. This means that older software, particularly that written in less common or outdated programming languages, might be underrepresented. Additionally, software with higher code quality and better documentation is more likely to be made readily available and thus may have been selected more frequently. This selection process could introduce bias in the distribution of models. Specifically, the simulation model distribution does not favour certain sectors. For instance, only 2 out of the 18 global biomes impact models were readily available and therefore included in our assessment. This may affect the generalizability of our findings across different domains of Earth System Sciences.

Moreover, our sustainability indicators do not cover other relevant aspects of sustainable research software, such as user base size, code development activity (e.g. frequency of code contributions, date of last update or version), number of publications and citations, coupling and cohesion, information content of comments, software adaptability to user requirement and interoperability. A larger user base often results in more reported bugs, which ultimately enhances software reliability. However, determining the exact size of the user base presents challenges due to data reliability issues. Additionally, there is the question of whether to include model output (data) users as part of the user base. Code development activity, such as the frequency of code contributions, indicates an ongoing commitment to improving and maintaining the software, but it does not necessarily reflect the quality of those contributions. In addition, the date of the last update or version is a useful metric, but it can be complex to interpret. For instance, research software might have an old last update date but still be widely used and reliable. Hence, these metrics were not evaluated here. The number of publications and citations referencing a model serves as an indicator of its impact and relevance within the research community. Yet, collecting and analysing this data is a time-consuming and complex task. We further did not evaluate the interdependence of software modules (coupling) and how functions in a module work towards the purpose of the module (cohesion) (Sarkar et al., 2008), as language-specific tools are required to evaluate such properties.

In addition to the previously discussed limitations, the indicator analysed in this study are quantitative metrics that can be measured. Factors such as information content of comments, software adaptability to user requirements and interoperability (Chue Hong et al., 2022) are examples of qualitative metrics that contribute to software sustainability. However, qualitative analysis is outside the scope of this study. We focus on measurable metrics that can be easily applied by the scientific community and by novice developers.

Also, we did not explore the analysis of code compliance to standards for other programming languages used for GIM development. Specifically for Python, the Pylint tool provides a lint score for all source code analysed, making it easier to interpret results. However, the tools for other languages (e.g., lintr for R) does not have this feature, which presents challenges in result interpretation.

Furthermore, future research could compare the sustainability levels of impact models developed by professional software design teams with those created in academic settings by non-professional software developers.

## 5 Recommendations

Making our research software sustainable requires a combined effort of the modelling community, scientific publishers, funders, and academic and research organizations that employ modelling researchers (Barker et al., 2022; Barton et al., 2022; McKiernan et al., 2023; Research Software Alliance, 2023). Some scientific publishers, research organizations, funders and scientific communities adopted and proposed solutions to this challenge, such as 1) requiring that authors make source code and workflows available, 2) implementing FAIR standards, 3) providing training and certification programs in software

engineering and reproducible computational research, 4) providing specific funding for sustainable software development, 5) establishing the support of permanently employed research software engineers for disciplinary software developers and 6) recognizing the scientific merit of sustainable research software by acknowledging and rewarding the development of high-quality, sustainable software as valuable scientific output in evaluation, hiring, promotions, etc. (Carver et al., 2022; Döll et al., 2023; Editorial, 2018; Eeuwijk et al., 2021; Merow et al., 2023). This software should be treated as a citable academic

contribution and included, for instance, in PhD theses (Merow et al., 2023).

To assess the current state of these practices in Earth system science, we conducted an analysis of sustainability indicators across global impact models. Our findings reveal that while some best practices are widely adopted, others are significantly lacking. Specifically, we found high implementation rates for documentation, open-source licensing, version control, and active developer involvement. However, four out of eight sustainability indicators showed poor implementation: automated

testing suites, containerization, sufficient comment density, and modularity. Additionally, only 50% of Python-specific models adhere to Python-based coding standards. These results highlight the urgent need for improved software development practices in Earth system science. Based on the results of our study, as well as the findings from existing literature, we propose the following actionable best practices for researchers developing software (summarized in Fig. 9):

• *Choose project management practices that align with your institutional environment, culture, and project requirements:* This can help plan, organize, and monitor your software development process, as well as improve collaboration and communication within your team and with stakeholders. Project management practices also help identify and mitigate risks, manage changes, and deliver quality software on time and within budget (Anzt et al., 2021). While traditional methods may be better suited for projects with fixed requirements, certain principles from

more flexible frameworks, such as Agile, can provide benefits in environments where requirements evolve or adaptability is critical. For example, Agile's iterative approach allows for incorporating changing research questions and hence software modifications or extensions, improving responsiveness to new developments (Turk et al., 2005).

- *Consider software architecture (organisation of software components) and requirements (user needs):* This will help design your software in a way that meets the needs and expectations of your users. Considering software architecture (such as Model-Controller-View (Guaman et al., 2021)) and user requirements helps to design a software system that has a clear and coherent structure, well-defined functionality, and suitable quality (Jay and Haines, 2019).

- *Select an open-source license:* Choosing an open-source license will make your software accessible and open to the research community, enable collaborations with other developers and contributors, as well as protect your intellectual property rights (Anzt et al., 2021; Carver et al., 2022). Accessible software is crucial to reduce reliance on email requests (Barton et al., 2022).

- *Use version control:* Version control can help you track and manage changes to your source code, which ensures the traceability of your software and facilitates reproducibility of scientific results generated by all prior versions of the software (Jiménez et al., 2017). Platforms like GitHub and GitLab are commonly used for this purpose. However, it's important to note that these platforms are not archival - the code can be removed by the developer at any time. A current best practice is to use both GitHub and GitLab for development, and to archive major releases on Zenodo or another archival repository.

- *Use coding standards accepted by your community (e.g., PEP8 for Python), good and consistent variable names, design principles, code quality metrics, peer code review, linters and software testing:* Coding standards help you write clear, consistent, and readable code that follows the best practices of your programming language and domain. It is key that developers consistently follow a coding style recognized by the relevant language community. Good variable names are descriptive and meaningful, reflecting the role and value of the variable. Design principles help adhere to the principles of sustainable research software, such as modularity, reusability and interoperability. These principles also guide the design of software by determining, for instance, the interaction of classes addressing aspects such as separation of concerns, abstraction, and encapsulation (Plösch et al., 2016).

  Code quality metrics can help measure and improve the quality of source code in terms of readability, maintainability, reliability, modularity and reusability. (Stamelos et al., 2002). Peer code review and linters (tools that analyse source code for potential errors) can help detect and fix errors, and vulnerabilities in your code, as well as improve your coding skills and knowledge (Jay and Haines, 2019). Software testing verifies if the research software performs as intended.

- *Make internal and external documentation comprehensible*: This can help you explain the purpose, functionality, structure, design, usage, installation, deployment, and maintenance of your software to yourself and others. Internal documentation refers to the comments and annotations within your code that describe what the code does and how it works. External documentation refers to manuals, guides, tutorials and any material that provide information about your software to users and developers. Comprehensible documentation can help you make your software more understandable, maintainable, and reusable. (Barker et al., 2022; Carver et al., 2022; Jay and Haines, 2019; Reinecke et al., 2022; Wilson et al., 2014)

- *Engage the research software community in the software development process.* This will help you get feedback, support, advice, collaboration, contribution and recognition from other researchers and developers who share your interests and goals. Engaging the research software community via conferences and workshops can also help you disseminate your software to a wider audience, increase its impact and visibility, and foster open science practices (Anzt et al., 2021). Additionally, consider utilizing containerization technologies, such as Docker, to simplify the installation and usage of your software (Nüst et al., 2020). It helps eliminate the "it works on my machine" problem. This approach also facilitates easy sharing of your software with software users. Furthermore, implement continuous integration and automated testing to maintain the quality and reliability of your code (Ståhl and Bosch, 2014). Continuous integration merges code changes from contributing developers frequently and automatically into a shared repository.

- *Integrate automation in development practices.* Automation plays a key role in streamlining software development by reducing manual effort and ensuring consistency (Wijendra and Hewagamage, 2021). We encourage developers to integrate automation into their workflows to improve efficiency. For instance, developers can use GitHub Actions to automate various tasks like running test suites, generating documentation, ensuring adherence to coding standards, and managing dependencies.

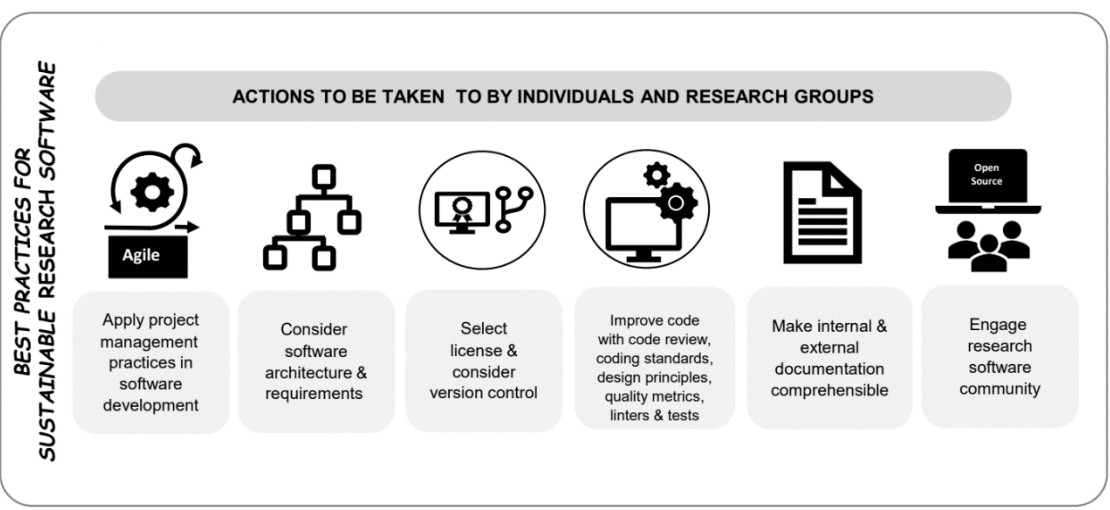

**Figure 9:** Actionable best practices for sustainable research software. The image summarizes the actions that modelling communities and individual developers should take, such as following project management practices, coding standards, reviews, documentation and community engagement strategies. These actions can help produce high-quality, robust, and reusable software that can be maintained.

## 6 Conclusion

The studied Earth system models are valuable and complex research tools that exhibit strengths and weaknesses in the use of certain software engineering practices (strengths, for example, in version control, open-source licensing, and documentation). However, notable areas remain for improvement, particularly in areas such as containerization and factors affecting code quality like comment density, modularity, and the availability of test suites. These shortcomings hinder the sustainability of such research software; they limit research reliability, reproducibility, collaboration, and scientific progress. To address this

challenge, we urge all stakeholders, such as scientific publishers, funders, as well as academic and research organizations, to facilitate the development and maintenance of sustainable research software. We also propose to use best practices for the developers of research software such as using project management and software design techniques, coding reviews, documentation, and community engagement strategies. We further suggest reprogramming the legacy code of well-established models. These practices can help achieve higher-quality code that is more understandable, reusable, and maintainable.

Efficient computational science requires high-quality software. While our study primarily focuses on Earth System Sciences, our assessment method and recommendations should be applicable to other scientific domains that employ complex research software. Future research could explore additional sustainability indicators, such as user base size, code development activity (e.g. frequency of code contributions), software adaptability and interoperability, as well as code compliance standards for various programming languages.

## Code Availability

The Python scripts utilized for analysis can be accessed at https://zenodo.org/doi/10.5281/zenodo.10245636 . Additionally, the line counting tool developed by Ben Boyter is available through the GitHub repository: https://github.com/boyter/scc.

## Data Availability

The results obtained from the line count analysis are accessible at https://zenodo.org/doi/10.5281/zenodo.10245636.

For convenient downloads of global impact models, links to the 32 global impact models, along with the respective dates of access, can be found in an Excel sheet named "ISIMIP_models.xlsx." present in the Zenodo repository.

## Author contributions

EN and RR designed the study. EN performed the analysis and wrote the paper with significant contributions from PD, DK, and RR. RR and PD supervised EN.

## Competing interests

The authors declare no competing interests.

## Acknowledgements

EN, RR, and PD acknowledge support from Deutsche Forschungsgemeinschaft (DFG) (Project number 443183317)

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
