# Peer review of "Software sustainability of global impact models"

_Geoscientific Model Development, 2024_

## Referee Comment (RC1)

**Review of Software Sustainability of global impact models**

Paper by Nyenah et al.

Review by Rolf Hut

The authors set out to study how sustainable (the software of) a collection of global impact models is and to provide the readership with hands-on advice on how to improve software sustainability. This work is an important addition to our (meta-)knowledge about the (software behind the) models that we use. I fits the scope of the journal and is well written.

I do, however, want to raise some points that I would invite the authors to respond to. Most of these points are more starters for discussion than concrete suggestions for how to change the paper.

**Major (including philosophical) point**

**On purpose of the work**

The work hinges on two thoughts: on the one hand do the authors conduct a study into the state of software sustainability and present their findings. This is a subject fitting of a research paper. On the other hand the authors write an opinion paper on how they advise the community to improve the sustainability of their software. While not made explicit in the paper, it seems (to me) that the authors use the analysis to justify the need for the advice given, for example: "in software development a best practice is to write (good) documentation. From our analysis we conclude that good documentation is lacking for X % of the GIMs studied. We therefore urgently advice the community to write better documentation.". Although not made explicit in the article, I think this is a good logical structure for an opinion article like this. My main concern is that of the six advice given three (maybe four) are covered by the questions and results of the analysis. The need for the other advice is not justified by the analysis.

- "apply project management" → not covered by survey (and see 'on agile' below).
- "consider software architecture" → maybe covered by analyses of lines of code per file, but hard to justify given different software architectures
- "select license & consider version control" → covered by analysis
- "improve code standards / test" → partially covered by analysis
- "make documentation comprehensible" → partially covered by analysis: quality is not checked, only if documentation is available at all.
- "Engage community" → marginally covered by analysis of active number of developers.

For more clarity and overall a stronger point towards the community I would restructure the paper to follow this logic:

1. We know that these advices are good practices in software design. (Make clear if the source of this best practice is literature or the lived experience of the authors themselves.)
2. To pinpoint the need for these advices we conduct an analysis that studies KPIs that signal of these advices are already implemented
3. For 3 out of 6 of these advices we find evidence that these are not being implemented
4. For the other advices, based on our lived experience, we think those are not being implemented
5. Science would be better if they were implemented broadly.

(note: this is an advice on how I would restructure the paper, I am looking forward to the reply of the authors with potential different viewpoints and I certainly do not say that restructuring like suggested is a "must" before the paper can be published).

**On purpose of research software**

The authors state that impact model "provide crucial information for policymakers, scientists and citizens" (line 38). Further on they argue that "research software that suffers from these shortcomings [….] impede research progress, decrease research efficiency and hinder scientific progress" (line 63). The citizens and policymakers seem to be out of the picture here. The point I want to raise is that different GIMs are made for different audiences. Most GIMs are made by scientists for scientists. The output of the model runs, the conclusions of the papers, can impact policymakers, but the model itself is not intended to be run, or analysed, by them. This contrasts with a number of GIMs that are used by policymakers (and / or citizens) themselves. For example, in my field (hydrology) the PCRGlobWB model (one of the GIMs in this paper) made by Utrecht University is used in academic research. On the other hand, the WFLOW model made by Deltares is both used by Deltares in consultancy work, it is run by operational institutes (governments) as part of their daily operations and it is used in academic publications. Organisations like Deltares (and DHI & SMHI for exampale) have dedicated software design teams. I wonder if the software behind models like this are at a different quality level compared to purely academic software. I would like to ask the authors to reflect on this and what it means for the interpretation of their analysis results.

**On representativeness of the selected GIMs**

The authors acknowledge the bias introduced by the selection mechanism of GIMs for their analysis. I think that the implications of this bias stretch further than the considerations they authors give in their 'limitations' section. The amount of 'legacy code' differs greatly between different fields of science. Not long ago (I'm getting old…) it was perfectly acceptable to make software available through "please email me". These models are still used, even as GIMs. By using this selection, not only does that exclude some fields of science, but it also selects on
- The type of programming language used at the time
- The type of code quality and style of documentation

Because of this, I'm afraid that the analysis of the authors shows a "best case" situation of the state of software quality in the field, which makes their advice all the more urgent.

**On costs and rewards**

The authors conclude that the quality of research software would improve if their advice is followed and they also conclude that writing and maintaining complex research software is expensive (line 386). The obvious questions that the authors do not address are: why is this as it is and who should pay for the change? To answer the first question, I would argue that scientists do not have an incentive to make their software better. A Good paper with bad software gets you just as far and writing bad code is much faster than nicely documenting and structuring your work (I have been guilty of this myself). Just asking scientist to "do better" will not change if those that do better are not rewarded for these efforts (or, more negatively, those providing bad code punished with negative career outcomes). I invite the authors to reflect on this and provide their vision on how to arrive on a situation where researchers feel they have the time, the budget and the incentives to actually spend time on writing better research software.

**On agile**

One of the advices that the authors give and that I want to push back on is to "apply project management practices" and in particular "Agile" when building research software. The 'agile' framework for developing software originated in the start-up culture of silicon valley and it is laced with assumptions that the environment that one works in is that of an anglo-saxon, shareholder value driven, software start-up. The main idea of Agile as I distil it is for a team at any point in time to decide on the next action that optimizes the value of the software product (sic) most. The underlying assumption are:
- It is unclear what the final product should be, we develop as we go, requirements can change any moment
- We have an existing product and an existing user base (customers) that we can continuously test our improvements with
- The crucial limit is the amount of development time (cost).

Especially the first point is plainly not true in many scientific projects such as big EU projects where it is decided upfront what should be delivered and requirements are fixed, yet budgets are not (always). (This is partly why many software projects in government and big institutes always go over budget).

I hope the authors recognize that I strongly push back against advocating for agile and would be much happier with something akin to: "choose a project management practice that honors the boundary conditions of the institutional environment and culture you are part of". I'm looking forward to reading the authors view on this.

**Minor points**
**On reproducibility through platforms**

With the risk of sounding like a reviewer that is mainly pushing their own work: reproducibility does not, per se, have to be fixed by the model developer. There are a few 'model platforms' that have emerged in the last years that give people access to either models itself, or the

output of models, in a FAIR way. This is also a way in which the community as a whole can arrange for better (reproducible) software. Projects that spring to mind from my own field include ESMValTool, eWaterCycle (the one I work on), PAVICS-Hydro, Deltares FEWS. Do the authors agree that this is a way as a community to make research software more reproduceable?

**On researchers as software developers**

On line 54 the authors claim that "most of these researchers are self-taught software developers with little knowledge of software requirements.". The authors do not back up this claim by referencing literature that has studied this. While I do agree that this might have been true a few decades ago, I would argue that currently at least all scientists get some sort of programming classes in their education and those doing a MSc or similar in "our fields" get quite extensive programming classes. This leads to two questions to the authors:
- do they stand by their claim?
- More related to the conclusion and how to implement the advices that the authors give: what do they think the role of education in general and graduate level eduation in particular should be?

**On the software cost estimation**

While I recognize the need to somehow quantify the amount of work (money) invested in I strongly think that the number of decimals in the coefficients presented hint at more certainty in the relation than the underlying data that this is fitted to justifies. I would like to ask the authors to do a bit of a sensitivity analyses on these coefficients to see how much their conclusions change. Furthermore, I think that the 18 large NASA projects include those kind of projects where an error in a line of code could mean that the lander misses the moon and astronauts die, where-as an error in a GIM usually means that a future projection is slightly off. (I'm being mean, but I do think that NASA spends more time on code quality control than most post-docs working on GIMs). I'd like to ask the authors to reflect on this a little bit in their discussion.

**Really minor points**
- Line 44: are non-physical impact models also possible (economical models? Demographic models? Statistical models?)
- Line 81: what about the cost of re-implementing / making reproducible versus the cost of maintaining old code?
- Line 102: I find it somewhat funny that the list of isimip models is itself not a FAIR datasource (no further action required)
- Line 165: could the absence of active developers not also imply very mature software that is just perfect?
- Line 171: singularity is now called apptainer
- Line 230: the references to sloc look weird: is this the preferred format?
- Table 2: I suggest to use '+' instead of 'x' to indicate availability.

- Figure 2: I suggest to add "no license" and "not OSI license" as columns to this figure.
- Figure 5 and figure 6: is it somehow possible to indicate which of these models are in which programming language? For example sort along the x-axis by programing language such that all the python models are next to each other?

---

## Referee Comment (RC2)

**Review of "Software Sustainability of global impact models"**

Facundo Fabián Sapienza
Monday 26th of August 2024

The manuscript contributes to the scientific literature by quantifying the level of software sustainability of different climate change impact models. This is done by first defining a series of nine indicators that evaluate factors such as automation, documentation, testing, and good coding practices present in the model.

I recommend the publication of the manuscript in GMD after the revisions and comments here presented are addressed. I believe this work brings light into the robustness, reproducibility and overall health of the software that we scientist develop. I personally enjoyed the reading of the manuscript, and I would be glad of seeing this work published. However, before publication I strongly suggest the authors to review and address the following major and minor points. If well overall the manuscript is easy to read and follow, some parts are a bit vague and required more explanations and/or more bibliographical references.

All comments, both major and minors, are aimed to improve the quality of the paper or to dilucidated some of my doubts or questions as I was reading the manuscript.

**Major Comments**

- It will help to the discussion the early introduction of one or more examples of software that follows the guidelines of "sustainable research software". I suggest using standard examples for this, for example a major library in Python (numpy, pandas, sklearn) with a link to their respective source code would help the reader to have an idea of how good software looks like.

- [146] **Version control.** My understanding is that in the manuscript the authors assess whether the models use or use not git or a similar version control system. However, I think it is also important to evaluate how version control has been used during the development of the different versions of the model. Did the contributors follow good version control practices (modular commits, push requests, discussions, versioning, etc), or simply put the final version of the software in GitHub or similar? I understand this is a finer analysis which I don't think is necessary to perform in the manuscript, but I would suggest mentioning this in the Version Control section since it is an important point.

- [154] **Use of open source license**. I think this section requires more discussion or at least a few references supporting the statement *"Open-source licenses foster collaboration and transparency by enabling community contributions and ensuring that software remains freely accessible"*. I partially agree with this statement, but I would like to see the why of this. Furthermore, I think is also important to mention the difference between different types of major licenses (copyleft or permissive), since I think this also has an impact in the meaning of the sentence. This point pops up every time the topic of licenses is addressed in the paper, so I strongly suggest the authors to address what do they mean by open-source licenses and how they relate to copyleft/permissive licenses.

- [159] **Number of active developers.** I think the number of active developers is a good proxy for evaluating how robust is the development of the software, but I don't think is the only one. I think with the same philosophy one can evaluate the number of commits, push requests, open and closed issues, etc. I suggest mentioning this in the manuscript, and maybe changing the "number of active developments" tag to something more generic that includes these other proxies or a more general concept (e.g., "software robustness"). Furthermore, it may be important to emphasize the distinction between developers and contributors: one can contribute to a project without writing

source code, but for example opening issues, managing a project, writing the documentation, etc. In this sense, contributors help to improve the robustness of the software, even when they don't directly write any line of code.

- [166] **Containerization.** I will suggest here making also a reference to cloud supported containers, such as Binder or GoogleColab, that allow the re-execution of the software. Under the hood, this also work as a container, but I think the deeper concept here is the capacity of re-executing an model with the computational environment requirements. This further resonates with the concept of analysis-ready data, cloud-optimized formats (see "Cloud-Native Repositories for Big Scientific Data" by Abernathey et. al, 2021) in the case of datasets. I think the same ideas apply here for models.

- [178] **Public availability of an (automated) testing suite.** I really like this point. I this is a good idea to look for automation of the tests. However, I would like to point out here that the concept of "automation" is in principle independent and complementary to the existence of the testing suite. Furthermore, the concept of automation applies also to some other indicators. For example, one can automate the creation of documentation using GitHub Pages, using GitHub actions to ensure the containerization of the software (even create all the ingredients for a docker file), and even ensure the compliance with coding standards of the software (see for example the Julia code formatter action: https://github.com/julia-actions/julia-format and I imagine there must be a way to automate the use of Pylint with Python).

- [269] How well documented are the models that just have a README file? In my experience, README documentations tend to be very plain and difficult to navigate. I think it is important to mention something about the quality of the documentation, at least as an observation.

- [295] Following my previous comment, I think it is important to state why licensing is important and the difference between licenses. References or further support here is needed. Copyleft and permissive licenses are very different. I would also suggest pointing which one of the licenses in Figure 2 are copyleft of permissive.

- [378] I don't understand the purpose of this section. The effort here is calculated using equation 1, which (besides being based in many assumptions that the authors do take care of) already suggested that more lines of code mean more effort, which is an expected concussion. With this point in mind, I don't fully understand what new conclusion are made in Section 3.3. There is a chance that here I am missing an important point, and in that case, I would like the authors to clarify. I think here it would be interesting to see how the effort correlated with the number of satisfied indicators. Without further analysis, I would suggest removing or perform a different analysis in this section.

- [410] I think this is a very important question and should be first raised in lines 197-200, and maybe postponing the discussion until later. It is not clear for me that 30-60% is the desired number, since most of the time other heuristics are used for determining the number of comments. In a nutshell, better code requires minimal commenting and clarity in the code. (See book "Beautiful Code" for a nice collection of essays around this precise point.) Furthermore, I think is important to mention the entanglement between writing good documentation and commenting, since many software documentations are generated automatically based on the comments in the code (e.g., based on the docstrings in Python and Julia).

- [490] Here it is mentioned a very important point: design principles, or design patterns. More than writing code that follows certain standards, it is important to think about the overall software architecture of the model. E.g., if I am working in Python, what are going to be my classes? How do they interact with each other? How data will be processed? Etc. None of the indicators really

address this aspect of software development (which I think is fine for the scope of the manuscript), but I think the authors should emphasize this point in the recommendation section.

- [496] As it was already mentioned, I think is important to remark that what here is referred as internal and external documentation sometimes are the same. Documentation can be created from comments in the code, specially docstrings, and this is actually a great documentation practice, to make the internal and external documentation to be the same so there is no repetition nor contradiction between the two.

- Since automation is mentioned as playing an important role, I would suggest including automation in a more general sense in the recommendation section. This includes using GitHub Actions to run the test suite, automate the creation of the documentation (static or dynamic), check for code style, check packages dependencies, etc. Another interesting tool the authors may want to consider mentioning is Makefile.

- Since the paper addresses the important aspect or sustainable software, I would strongly suggest that the code used to generate the analysis, and the figures of the manuscript are presented in the same standard that the nine indicators dictate. I think this will really improve the quality of the work, and it can serve as an example in the manuscript itself of "how things should be done". I think readers would like to see that, and I think it is part of the philosophy of the manuscript to promote these good practices. Furthermore, I would make the point that the same tools that had been used in this manuscript can be used for analyzing other source code models, so the code used in the manuscript can be re-usable by other users.

**Minor Comments**

- [30] I am a bit confused by what Earth system modelling entitles. When presenting the models, we are talking about models in agriculture, biomes, fire, etc. In line 33, it says that "While so-called Earth System Models always include the simulation of atmospheric processes and thus compute climate variables and how they change due to greenhouse gas emissions [...]". However, this excludes many "Earth system models", including all those not based in atmospheric processes. I think the phrasing of Earth system model should be narrowed to what the models are for. For example, all the models in the ISIMIP are about the impact of climate change.

- [66-70] Repeats reference Anzt et. al. (2021)

- [83-99] I suggest splitting this paragraph into two, with the second stating what is done specifically in this work (maybe just break before "In this study, we assess...").

- [105] Data in this line means "model"? I will suggest not to use "data" to refer to the models in the ISIMIP database, since it is a bit confusing. If this is referring to another type of data, maybe explain.

- [110-111] This sentence requires rephasing, since it is ambiguous what it means by "in the described way" (meaning GitHub/GitLab or also including source code in reference papers).

- [189] Is PEP8 the de-facto coding style in Python? I think there used to be some alternatives and may be important to mention something like this. Furthermore, in other programming languages there are more than one coding style that are accepted by the community (e.g., in Julia), so it may be important to mention that the important aspect of this is that the developers of one model stick with one style, rather that sticking with one single style.

- Table 1. Following the logic in the text, I will suggest making the division in the table between best practice in software engineering and source code quality (e.g., adding this information in the table, dividing them with a horizontal line). I think this will make the conceptual difference clearer than using the footnote. Check punctuation at the end of the descriptions. Some of the items end with dot, not dot, or comma.

- [245] These are the 32 models mentioned in line 110, right? I would mention this again for clarity.

- Figure 1. I will suggest sorting the bars in increasing/decreasing order. This is a comment I had with all the rest of the tables of the paper, where I would order the bars for clarity.

- [255] I will suggest not starting the sentence with a numeral since this is uncommon and non-recommended in formal English.

- [289] There is some repetition between what is said here and the paragraph in line 255 and Figure 1. I think mentioning git and the corresponding platform (GitHub, ...) should be made once in the same section for clarity in the text.

- Consider introducing Table 2 before Figure 1, since it contains the overall information of the model.

- It would be great to add an extra column to Table 2 with the year of the model (last version) and sort by this. I think it would be interesting to see if the availability of documentation, version control, etc, had improved over the years. Also interesting to see programming language used and how this changed over the years.

- [291] I don't agree with the statement that "Developers' preference for Git highlights its user-friendly nature and effectiveness in supporting collaborative efforts". I think there may be other reasons for this, since there were and are other version control systems that are as user-friendly as git but didn't became that popular. One reason for this is that GitHub naturally uses git, and that many developers use VSCode with also supports git and GitHub. If the authors want to keep this sentence as it is, the statement should be supported by references.

- [306] It is important to start saying what is a good number of contributors, or what is expected to see here. It is unclear to me that ~10 contributors is robust enough.

- [320] Mention which container platform these 5 models used. Did all used Docker? If so, how do they share it?

- [326] I am curious: do the models with test suite use a preferred programming language? Does the programming language plays a role in how easy is to implement the test suite? Maybe the authors want to answer to this question in the manuscript, I think it will make an interesting point.

- [249] I will suggest not starting the sentence with a numeral.

- [429] I really enjoyed this section, and I think it will improve to the communication of the paper to put this section earlier in the manuscript, maybe between the introduction of the indicators and the analysis and the results.

- [478] Same point about permissive and copyleft licenses. What do the authors mean by open-source licenses? Do you mean permissive?

- [482] I am not sure that version control ensures software reproducibility, not with other important tools (environment or containerization, testing suite, etc).

**General comments**

- I strongly suggest sorting all the tables in decreasing or increasing order for readability.

- I suggest the authors to do a English style revision of the manuscript since there are different styles in the manuscript. This includes the starting of the sentences with numerals and the use or not use of contractions.

- In the same style that Figure 9, I think a figure at the beginning of the manuscript summarizing the indicators and what are good software practices will improve the manuscript.

---

## Author Comment (AC1)

**Reply to Reviewer 1:  Rolf Hut**

Dear Reviewer,

We appreciate your prompt and critical review of our paper. Your thoughtful comments and suggestions have greatly improved the quality of our revised manuscript.

In the following sections, we have addressed your comments point-by-point and changed the manuscript accordingly. Please find the tracked changes attached to this letter. Your suggestions are highlighted in blue, while our responses are in black and new text italics. All section and line numbers mentioned correspond to the revised manuscript. In summary, we have revised aspects of the introduction and enhanced the results and discussion sections.

The authors set out to study how sustainable (the software of) a collection of global impact models is and to provide the readership with hands-on advice on how to improve software sustainability. This work is an important addition to our (meta-) knowledge about the (software behind the) models that we use. It fits the scope of the journal and is well-written.

Thank you for highlighting the importance and quality of our paper.

I do, however, want to raise some points that I would invite the authors to respond to. Most of these points are more starters for discussion than concrete suggestions for how to change the paper.

**Major (including philosophical) point**

**On purpose of the work**

The work hinges on two thoughts: on the one hand do the authors conduct a study into the state of software sustainability and present their findings. This is a subject fitting of a research paper. On the other hand the authors write an opinion paper on how they advise the community to improve the sustainability of their software. While not made explicit in the paper, it seems (to me) that the authors use the analysis to justify the need for the advice given, for example: "in software development a best practice is to write (good) documentation. From our analysis we conclude that good documentation is lacking for X % of the GIMs studied. We therefore urgently advice the community to write better documentation.". Although not made explicit in the article, I think this is a good logical structure for an opinion article like this. My main concern is that of the six advice given three (maybe four) are covered by the questions and results of the analysis. The need for the other advice is not justified by the analysis.

- "apply project management" not covered by survey (and see 'on agile' below).
- "consider software architecture" maybe covered by analyses of lines of code per file, but hard to justify given different software architectures
- "select license & consider version control" covered by analysis
- "improve code standards / test" partially covered by analysis
- "make documentation comprehensible" partially covered by analysis: quality is not checked, only if documentation is available at all.
- "Engage community" marginally covered by analysis of active number of developers.

For more clarity and overall a stronger point towards the community I would restructure the paper to follow this logic:

1. We know that these advices are good practices in software design. (Make clear if the source of this best practice is literature or the lived experience of the authors themselves.)
2. To pinpoint the need for these advices we conduct an analysis that studies KPIs that signal of these advices are already implemented
3. For 3 out of 6 of these advices we find evidence that these are not being implemented
4. For the other advices, based on our lived experience, we think those are not being implemented
5. Science would be better if they were implemented broadly.

(note: this is an advice on how I would restructure the paper, I am looking forward to the reply of the authors with potential different viewpoints and I certainly do not say that restructuring like suggested is a "must" before the paper can be published).

Thank you very much for the valuable feedback. We agree that the paper in its previous version was a balancing act between evidence and opinion in the last section of the paper. Even though we did not restructure the whole paper, we revised our recommendation section by linking them more clearly to the core findings of our paper. Furthermore, we have now expressed more clearly that the recommendations that cannot be backed up with our own results are based on other findings in the literature. The section now reads (Section 5, Line 495-513)

*"Making our research software sustainable requires a combined effort of the modelling community, scientific publishers, funders, and academic and research organizations that employ modelling researchers (Barker et al., 2022; Barton et al., 2022; McKiernan et al., 2023; Research Software Alliance, 2023). Some scientific publishers, research organizations, funders and scientific communities adopted and proposed solutions to this challenge, such as 1) requiring that authors make source code and workflows available, 2) implementing FAIR standards, 3) providing training and certification programs in software engineering and reproducible computational research, 4) providing specific funding for sustainable software development, 5) establishing the support of permanently employed research software engineers for disciplinary software developers and 6) recognizing the scientific merit of sustainable research software by acknowledging and rewarding the development of high-quality, sustainable software as valuable scientific output in evaluation, hiring, promotions, etc. (Carver et al., 2022; Döll et al., 2023; Editorial, 2018; Eeuwijk et al., 2021; Merow et al., 2023). This software should be treated as a citable academic contribution and included, for instance, in PhD theses (Merow et al., 2023).*

*To assess the current state of these practices in Earth system science, we conducted an analysis of sustainability indicators across global impact models. Our findings reveal that while some best practices are widely adopted, others are significantly lacking. Specifically, we found high implementation rates for documentation, open-source licensing, version control, and active developer involvement. However, four out of eight sustainability indicators showed poor implementation: automated testing suites, containerization, sufficient comment density, and modularity. Additionally, only 50% of Python-specific models adhere to Python-based coding standards. These results highlight the urgent need for improved software development practices in Earth system science. Based on the results of our study, as well as the findings from existing literature, we propose the following actionable best practices for researchers developing software (summarized in Fig. 9):"*

**On purpose of research software**

The authors state that impact model "provide crucial information for policymakers, scientists and citizens" (line 38). Further on they argue that "research software that suffers from these shortcomings [....] impede research progress, decrease research efficiency and hinder scientific progress" (line 63). The citizens and policymakers seem to be out of the picture here. The point I want to raise is that different GIMs are made for different audiences. Most GIMs are made by scientists for scientists. The output of the model runs, the conclusions of the papers, can impact policymakers, but the model itself is not intended to be run, or analysed, by them. This contrasts with a number of GIMs that are used by policymakers (and / or citizens) themselves. For example, in my field (hydrology) the PCRGlobWB model (one of the GIMs in this paper) made by Utrecht University is used in academic research. On the other hand, the WFLOW model made by Deltares is both used by Deltares in consultancy work, it is run by operational institutes (governments) as part of their daily operations and it is used in academic publications. Organisations like Deltares (and DHI & SMHI for example) have dedicated software design teams. I wonder if the software behind models like this are at a different quality level compared to purely academic software. I would like to ask the authors to reflect on this and what it means for the interpretation of their analysis results.

While we acknowledge that different models are created for different purposes, we set out to specifically understand global impact models in this paper. Global Impact Models (GIMs) are created for scientific applications; the key issue we aim to highlight is the potential limitation for our scientific insights resulting from unsustainable GIMs, rather than the technical aspects of running these models. The outputs of such models—whether data, publications, or reports—are, however, very much used or at least noticed by policymakers and citizens and impact their decision-making. Therefore, potential errors in these outputs can misinform decision-making processes.

1) To clarify this, we have revised our statement about GIMs in the introduction to: (Section 1, Line 35-38)

"These impact models also quantify the historical development and current situation of key environmental issues such as water stress, wildfire hazard, and fish population. The outputs of these models whether data, publications or reports thus provide crucial information for policymakers, scientists, and citizens."

2) Although comparing the sustainability levels of research software developed in academia (by Ph.D. students and postdoctoral researchers) with that created by dedicated software design teams is an intriguing idea, it would lead to a different study setup. These software rarely exist as global scale models (except maybe the SMHI Hype, which, however, does not participate in ISIMIP), which is the focus of this paper. For example, the suggested WFLOW model is a catchment-scale hydrological model and thus falls outside the scope of this paper. We acknowledge, however, the value of this suggestion, and we refine our limitation sections to include this.

(Section 4, Line 492-493) *"Furthermore, future research could compare the sustainability levels of impact models developed by professional software design teams with those created in academic settings by non-professional software developers."*

**On representativeness of the selected GIMs**

The authors acknowledge the bias introduced by the selection mechanism of GIMs for their analysis. I think that the implications of this bias stretch further than the considerations they authors give in

their 'limitations' section. The amount of 'legacy code' differs greatly between different fields of science. Not long ago (I'm getting old…) it was perfectly acceptable to make software available through "please email me". These models are still used, even as GIMs. By using this selection, not only does that exclude some fields of science, but it also selects on

- The type of programming language used at the time
- The type of code quality and style of documentation

Because of this, I'm afraid that the analysis of the authors shows a "best case" situation of the state of software quality in the field, which makes their advice all the more urgent.

We agree that the selection process may have introduced bias. We have revised our limitation section to include stated suggestions.

(Section 4, Line 462-468) "Our study has limitations in the following regards. In the interest of timely analysis, we did not contact the developers of models that were not readily available. This means that older software, particularly that written in less common or outdated programming languages, might be underrepresented. Additionally, software with higher code quality and better documentation is more likely to be made readily available and thus may have been selected more frequently. This selection process could introduce bias in the distribution of models. Specifically, the simulation model distribution does not favour certain sectors. For instance, only 2 out of the 18 global biomes impact models were readily available and therefore included in our assessment. This may affect the generalizability of our findings across different domains of Earth System Sciences."

**On costs and rewards**

The authors conclude that the quality of research software would improve if their advice is followed and they also conclude that writing and maintaining complex research software is expensive (line 386). The obvious questions that the authors do not address are: why is this as it is and who should pay for the change? To answer the first question, I would argue that scientists do not have an incentive to make their software better. A Good paper with bad software gets you just as far and writing bad code is much faster than nicely documenting and structuring your work (I have been guilty of this myself). Just asking scientist to "do better" will not change if those that do better are not rewarded for these efforts (or, more negatively, those providing bad code punished with negative career outcomes). I invite the authors to reflect on this and provide their vision on how to arrive on a situation where researchers feel they have the time, the budget and the incentives to actually spend time on writing better research software.

We agree with the reviewer and therefore revise line 386 based on your suggestion to:

(Section 3, Line 414-419) "*The results suggest that these complex research software programs are expensive tools that require adequate funding for development and maintenance to make them sustainable. This is consistent with previous studies that have highlighted funding challenges for developing and maintaining sustainable research software in various domains (Carver et al., 2013, 2022; Eeuwijk et al., 2021; Merow et al., 2023; Reinecke et al., 2022). Merow et al. (2023) also emphasized that the accuracy and reproducibility of scientific results increasingly depend on updating and maintaining software. However, the incentive structure in academia for software development — and especially maintenance — is insufficient (Merow et al., 2023).*"

Also, regarding a vision of producing better software and who should pay, we provide recommendations based on literature in the recommendation section (Section 5, Line 495-505), with which we already responded to your comment on the purpose of the work on page 2 of this response.

"Making our research software sustainable requires a combined effort of the modelling community, scientific publishers, funders, and academic and research organizations that employ modelling researchers (Barker et al., 2022; Barton et al., 2022; McKiernan et al., 2023; Research Software Alliance, 2023). Some scientific publishers, research organizations, funders and scientific communities adopted and proposed solutions to this challenge, such as 1) requiring that authors make source code and workflows available, 2) implementing FAIR standards, 3) providing training and certification programs in software engineering and reproducible computational research, 4) providing specific funding for sustainable software development, 5) establishing the support of permanently employed research software engineers for disciplinary software developers and 6) recognizing the scientific merit of sustainable research software by acknowledging and rewarding the development of high-quality, sustainable software as valuable scientific output in evaluation, hiring, promotions, etc. (Carver et al., 2022; Döll et al., 2023; Editorial, 2018; Eeuwijk et al., 2021; Merow et al., 2023). This software should be treated as a citable academic contribution and included, for instance, in PhD theses (Merow et al., 2023)."

**On agile**

One of the advices that the authors give and that I want to push back on is to "apply project management practices" and in particular "Agile" when building research software. The 'agile' framework for developing software originated in the start-up culture of silicon valley and it is laced with assumptions that the environment that one works in is that of an anglo-saxon, shareholder value driven, software start-up. The main idea of Agile as I distil it is for a team at any point in time to decide on the next action that optimizes the value of the software product (sic) most. The underlying assumption are:

- It is unclear what the final product should be, we develop as we go, requirements can change any moment
- We have an existing product and an existing user base (customers) that we can continuously test our improvements with
- The crucial limit is the amount of development time (cost).

Especially the first point is plainly not true in many scientific projects such as big EU projects where it is decided upfront what should be delivered and requirements are fixed, yet budgets are not (always). (This is partly why many software projects in government and big institutes always go over budget). I hope the authors recognize that I strongly push back against advocating for agile and would be much happier with something akin to: "choose a project management practice that honors the boundary conditions of the institutional environment and culture you are part of". I'm looking forward to reading the authors view on this.

We agree with the reviewer, particularly regarding the limitations of Agile in certain scientific settings, such as those with fixed requirements. However, we believe that some principles and practices from Agile can still provide significant value in academic software development, particularly in environments where timelines are constrained and requirements do change quickly (with newly available data or methods), such as PhD and Postdoc positions. Below, we outline some key benefits:

- Flexibility in Handling Evolving Requirements: While many scientific projects have well-defined requirements, others benefit from Agile's adaptability to changing conditions. For

example, as new research questions arise or additional factors need to be considered (e.g., integrating karst regions into hydrological research software for groundwater recharge), Agile's iterative process allows for the efficient incorporation of these updates. This flexibility ensures that the software remains responsive to ongoing developments.

- Transparency and Progress Tracking: Agile's use of tools like backlogs, task boards, and regular progress updates provides clear visibility into the project's status. This is particularly useful in academic settings, where project continuity can be disrupted due to personnel changes, such as the arrival or departure of researchers. These tools help ensure smooth transitions by clearly indicating completed tasks and those still in progress, thereby minimizing disruptions to the workflow.
- Enhanced Collaboration and Communication: Agile emphasizes frequent communication among team members and stakeholders, which can be highly beneficial in academic settings. Regular meetings and updates help ensure alignment among diverse contributors, such as students, researchers, and supervisors. This ongoing collaboration helps keep the team informed and engaged, and allows for timely input and feedback as the project evolves.

To be more nuanced about our recommendation we have revised our recommendation on project management practices to

*(Section 5, Line 515-522) "Choose project management practices that align with your institutional environment, culture, and project requirements: This can help plan, organize, and monitor your software development process, as well as improve collaboration and communication within your team and with stakeholders. Project management practices also help identify and mitigate risks, manage changes, and deliver quality software on time and within budget (Anzt et al., 2021). While traditional methods may be better suited for projects with fixed requirements, certain principles from more flexible frameworks, such as Agile, can provide benefits in environments where requirements evolve or adaptability is critical. For example, Agile's iterative approach allows for incorporating changing research questions and hence software modifications or extensions, improving responsiveness to new developments (Turk et al., 2005)."*

**Minor points**

**On reproducibility through platforms**.

With the risk of sounding like a reviewer that is mainly pushing their own work: reproducibility does not, per se, have to be fixed by the model developer. There are a few 'model platforms' that have emerged in the last years that give people access to either models itself, or the output of models, in a FAIR way. This is also a way in which the community as a whole can arrange for better (reproducible) software. Projects that spring to mind from my own field include ESMValTool, eWaterCycle (the one I work on), PAVICS-Hydro, Deltares FEWS. Do the authors agree that this is a way as a community to make research software more reproduceable?

Thank you for your insightful comments on reproducibility through platforms. We agree that reproducibility is not solely the responsibility of the model developer. The emergence of community-driven model platforms like ESMValTool, eWaterCycle, PAVICS-Hydro, and Deltares FEWS provides a FAIR way to access models and their outputs. For example, eWaterCycle allows users to run hydrological models in containers, ensuring consistency and reproducibility (as we encourage developers to use, see Section 5: Line 561). Additionally, tools like ESMValTool eliminate the need to create custom code to assess model outputs (diagnostics), significantly enhancing reproducibility.

**On researchers as software developers**

On line 54 the authors claim that "most of these researchers are self-taught software developers with little knowledge of software requirements.". The authors do not back up this claim by referencing literature that has studied this. While I do agree that this might have been true a few decades ago, I w ould argue that currently at least all scientists get some sort of programming classes in their education and those doing a MSc or similar in "our fields" get quite extensive programming classes. This leads to two questions to the authors:

- do they stand by their claim?

We stand by our claim based on recent studies by Reinecke et al. (2022) and Nangia & Katz (2017). To provide clarity and support for our statement, we have revised it as follows:

(Section 1, Line 53-57) "Most of these researchers are self-taught software developers (Nangia and Katz, 2017; Reinecke et al., 2022) with little knowledge of software requirements (specifications and features of software), industry-standard software design patterns (Gamma et al., 1994), good coding practices (e.g., using descriptive variable names), version control, software documentation, automated testing and project management practice (e.g. agile) (Carver et al., 2013, 2022; Hannay et al., 2009; Reinecke et al., 2022)."

- More related to the conclusion and how to implement the advices that the authors give: what do they think the role of education in general and graduate level education in particular should be?

Education, especially at the graduate level, could focus on developing essential skills in software engineering and reproducible computational research in addition to other mention in Section 5 Line 495-505.

**On the software cost estimation**

While I recognize the need to somehow quantify the amount of work (money) invested in I strongly think that the number of decimals in the coefficients presented hint at more certainty in the relation than the underlying data that this is fitted to justifies. I would like to ask the authors to do a bit of a sensitivity analyses on these coefficients to see how much their conclusions change. Furthermore, I think that the 18 large NASA projects include those kind of projects where an error in a line of code could mean that the lander misses the moon and astronauts die, where-as an error in a GIM usually means that a future projection is slightly off. (I'm being mean, but I do think that NASA spends more time on code quality control than most post-docs working on GIMs). I'd like to ask the authors to reflect on this a little bit in their discussion.

We conducted a sensitivity analysis on the COCOMO model coefficients as requested. The results are now included as Supplementary Figure S2. With a small additive change (±0.1) in these coefficients, the estimated effort ranges from approximately 1 to 255 person-months on -0.1 scenario, and up to about 960 person-months on the +0.1 scenario(Supplementary Fig. S2). These findings generally show that these models require a lot of effort especially if the model has large total lines of code.

We revise our manuscript with these new findings.

(Section 3, Line 409-412) *"As the TLOC of the impact model codes ranges from 262 to 500,000 TLOC (Fig. 7), the effort required to produce these models ranges from 1 to 495 person-months (Fig. 7). With a small additive change of ±0.1 of the COCOMO model coefficients, the range of estimated effort changes to 1 to 255 person-months in the case of -0.1 , and to 1 to 960 person-months in the case of +0.1 (Supplementary Fig. S2)."*

Also, we acknowledge that likely NASA invests significantly more time and resources into code quality control due to the critical nature of their projects (however we do not know if that is really the case). While we do not evaluate or discuss NASA's quality control processes, as that is beyond the scope of this paper, we aim to provide a rough estimate of the cost of producing research software. We utilize cost model built on NASA's data in our paper because it offers a simplified alternative compared to other software cost models and is widely used. The NASA projects used in developing this cost model contain software components similar to research software, making it a suitable reference point for our estimation purposes.

**Really minor points**

- Line 44: are non-physical impact models also possible (economical models? Demographic models? Statistical models?)

    While impact models can indeed represent both physical and non-physical processes, such as economic or demographic models, our focus in this paper is solely on impact models in participating in ISIMIP.

- Line 81: what about the cost of re-implementing / making reproducible versus the cost of maintaining old code?

    Thank you for the comment. We do not address the cost of re-implementing or making code reproducible versus the cost of maintaining old code in this study, as it falls outside the scope of this study.
    We now state that explicitly in the revised manuscript (Section 1, Line 101-102)

    *"We further provide first-order cost estimates required to develop these GIMs but do not address the cost of re-implementing or making code reproducible versus the cost of maintaining old code in this study."*

- Line 102: I find it somewhat funny that the list of isimip models is itself not a FAIR datasource (no further action required)

    That is certainly true. We are thinking about presenting our findings to the ISIMIP community at one of their next conferences to discuss this.

- Line 165: could the absence of active developers not also imply very mature software that is just perfect?

    While it is possible that the absence of active developers could suggest the software is very mature and stable, it is unlikely that software can ever be "perfect." In practice, users often request new features or bug fixes over time. As technologies, libraries, and hardware evolve, software typically needs updates to remain compatible and secure. Even highly mature software may require occasional maintenance to address security vulnerabilities, adapt to new operating systems, or improve performance. Therefore, the need for active developers is almost always present to ensure long-term relevance and usability.

- Line 171: singularity is now called apptainer
Thank you for the correction. We have revised the name accordingly. Section now reads (Section 2, Line 185-186) "Some popular containerization solutions include Docker (https://www.docker.com/) and Apptainer (https://apptainer.org/)."

- Line 230: the references to sloc look weird: is this the preferred format?
We have corrected the reference.

- Table 2: I suggest to use '+' instead of 'x' to indicate availability.
We have revised Table 2 accordingly.

- Figure 2: I suggest to add "no license" and "not OSI license" as columns to this figure.
In Figure 2, there is no column labelled 'not OSI license.' Instead, we present research software categorized by its licensing information: 25% of the software lacks any license information, while 6% use licenses that are not OSI-approved. This means these licenses do not conform to the Open Source Definition, which ensures that software can be freely used, modified, and shared. We also now refine our methodology section which clarifies the meaning of OSI-approved licenses.

(Section 2, Line 164-167) *"We determined the existence of open-source licenses by checking license files within repositories or official websites against licenses approved by the Open Source Initiative (OSI) (https://opensource.org/licenses). Specifically, we looked for licenses that conform to the Open Source Definition, which ensures that software can be freely used, modified, and shared (Colazo and Fang, 2009; Rashid et al., 2019)."*

- Figure 5 and figure 6: is it somehow possible to indicate which of these models are in which programming language? For example sort along the x-axis by programing language such that all the python models are next to each other?
Thank you for the suggestion, however sorting the models by programming language would disrupt the ability to compare performance across sectors, which is the main focus of Figures 5 and 6. To address the need for information about programming languages, we have updated Table 2 to include all relevant programming languages for each model.

[revised manuscript text omitted]

---

## Author Comment (AC2)

**Reply to Reviewer 2:  Facundo Fabián Sapienza**

Dear Reviewer,

We appreciate your prompt and critical review of our paper. Your thoughtful comments and suggestions have greatly improved the quality of our revised manuscript.

In the following sections, we have addressed your comments point-by-point and changed the manuscript accordingly. Please find the tracked changes attached to this letter. Your suggestions are highlighted in blue, while our responses are in black and new text italics. All section and line numbers mentioned correspond to the revised manuscript. In summary, we have revised aspects of the introduction and enhanced the results and discussion sections.

The manuscript contributes to the scientific literature by quantifying the level of software sustainability of different climate change impact models. This is done by first defining a series of nine indicators that evaluate factors such as automation, documentation, testing, and good coding practices present in the model.

I recommend the publication of the manuscript in GMD after the revisions and comments here presented are addressed. I believe this work brings light into the robustness, reproducibility and overall health of the software that we scientist develop. I personally enjoyed the reading of the manuscript, and I would be glad of seeing this work published. However, before publication I strongly suggest the authors to review and address the following major and minor points. If well overall the manuscript is easy to read and follow, some parts are a bit vague and required more explanations and/or more bibliographical references.

We appreciate the positive feedback on our work and have addressed the raised points in the following.

All comments, both major and minors, are aimed to improve the quality of the paper or to dilucidated some of my doubts or questions as I was reading the manuscript.

**Major Comments**

-   It will help to the discussion the early introduction of one or more examples of software that follows the guidelines of "sustainable research software". I suggest using standard examples for this, for example a major library in Python (numpy, pandas, sklearn) with a link to their respective source code would help the reader to have an idea of how good software looks like.

    Thank you for your valuable feedback. We have revised our introduction to include your suggestion.

    (Section 1, Line 68-74) *"There are various interpretations of the meaning of "sustainable research software". Anzt et al. (2021) define research software as software that is maintainable, extensible, flexible (adapts to user requirements), has a defined software architecture, is testable, has comprehensive in-code and external documentation, and is accessible (the software is licensed as Open Source with a digital object identifier (DOI) for proper attribution) (Anzt et al., 2021). For example, NumPy (https://numpy.org/) is a widely used scientific software package that exemplifies many of these qualities (Harris et al., 2020). Although NumPy is not an impact model, it is an exemplar of sustainable research software; it*

*is open-source, maintains rigorous version control and testing practices, and is extensively documented, making it highly reusable and extensible for the scientific community."*

- [146] **Version control**. My understanding is that in the manuscript the authors assess whether the models use or use not git or a similar version control system. However, I think it is also important to evaluate how version control has been used during the development of the different versions of the model. Did the contributors follow good version control practices (modular commits, push requests, discussions, versioning, etc), or simply put the final version of the software in GitHub or similar? I understand this is a finer analysis which I don't think is necessary to perform in the manuscript, but I would suggest mentioning this in the Version Control section since it is an important point.

Thank you for your thoughtful comment. We agree that evaluating the implementation of version control practices, such as modular commits, pull requests, discussions, and versioning would provide valuable insights and would be interesting for follow-up studies. In line with your suggestion, we have mentioned these practices in the revised section. The updated version now reads:

(Section 2, Line 153-161) *"Version control. Version control systems such as Git and Mercurial facilitate track changes, and collaborative development, and provide a history of software evolution. To assess whether GIMs use version control for development, we focused on commonly used open-source version control hosting repositories such as GitLab, GitHub, BitBucket, Google Code, and Source Forge. The hostname such as "github" or "gitlab" in the source link of models provides clear indications of version control adoption in their development process. For other models, we searched within the Google search engine using model names and keywords such as "Bitbucket", "Google Code", and "Source Forge". While we focus on identifying the use of version control systems, evaluating how version control was implemented during the development process — such as the use of modular commits, pull requests, discussions, and proper versioning — is a finer analysis that falls beyond the scope of this study. However, such practices are crucial for ensuring high-quality software development and collaborative practices."*

- [154] **Use of open source license**. I think this section requires more discussion or at least a few references supporting the statement "Open-source licenses foster collaboration and transparency by enabling community contributions and ensuring that software remains freely accessible". I partially agree with this statement, but I would like to see the why of this. Furthermore, I think is also important to mention the difference between different types of major licenses (copyleft or permissive), since I think this also has an impact in the meaning of the sentence. This point pops up every time the topic of licenses is addressed in the paper, so I strongly suggest the authors to address what do they mean by open-source licenses and how they relate to copyleft/permissive licenses.

Thank you for your insightful feedback. We have removed the statement regarding open-source licenses fostering collaboration and transparency, as we agree that it requires more discussion and supporting references. Additionally, we have expanded the section to include a detailed explanation of the key differences between copyleft and permissive licenses, the two major categories of open-source licenses. In our study, we focus solely on the presence of an open-source license regardless of the type of open source license.  The revised section now reads:

(Section 2, Line 164-172) *"Use of an open-source license. We determined the existence of open-source licenses by checking license files within repositories or official websites against licenses approved by the Open Source Initiative (OSI) (https://opensource.org/licenses). Specifically, we looked for licenses that conform to the Open Source Definition, which ensures that software can be freely used, modified, and shared (Colazo and Fang, 2009; Rashid et al., 2019). There are two major categories of open-source licenses: permissive licenses, such as MIT or Apache, that allow for minimal restrictions on how the software can be used (e.g., providing attribution), and copyleft licenses, like GPL, that require derivatives to maintain the same licensing terms (Colazo and Fang, 2009; Rashid et al., 2019). Although these licenses differ in their terms, both contribute to collaboration and transparency. In this study, we only check if the software is open-source, regardless of the type of open-source license."*

- [159] **Number of active developers**. I think the number of active developers is a good proxy for evaluating how robust is the development of the software, but I don't think is the only one. I think with the same philosophy one can evaluate the number of commits, push requests, open and closed issues, etc. I suggest mentioning this in the manuscript, and maybe changing the "number of active developments" tag to something more generic that includes these other proxies or a more general concept (e.g., "software robustness"). Furthermore, it may be important to emphasize the distinction between developers and contributors: one can contribute to a project without writing source code, but for example opening issues, managing a project, writing the documentation, etc. In this sense, contributors help to improve the robustness of the software, even when they don't directly write any line of code.

We agree that the number of active developers is not the only indicator if the goal is to measure the project's robustness, and we appreciate your suggestion to consider additional factors such as commits, pull requests, and open/closed issues. Our measure of active developers serves as an indicator of ongoing maintenance and the prevention of software stagnation, rather than a definitive measure of a project's robustness (see Section 2, Line 174). Therefore, we prefer to keep the term "number of active developers". Also, to avoid confusion between the terms "contributor" and "developer," we use "developer" exclusively, as contributors also write lines of code according to the definition from the Mozilla Public License (https://www.mozilla.org/en-US/MPL/2.0/).

- [166] **Containerization**. I will suggest here making also a reference to cloud supported containers, such as Binder or GoogleColab, that allow the re-execution of the software. Under the hood, this also work as a container, but I think the deeper concept here is the capacity of re-executing an model with the computational environment requirements. This further resonates with the concept of analysis-ready data, cloud-optimized formats (see "Cloud-Native Repositories for Big Scientific Data" by Abernathey et. al, 2021) in the case of datasets. I think the same ideas apply here for models.

Thank you for your thoughtful comment. We have incorporated your suggestions into the section and it now reads.

(Section 2, Line 185-189) *"Some popular containerization solutions include Docker (https://www.docker.com/) and Apptainer (https://apptainer.org/). There are also cloud-supported container solutions such as Binder (https://mybinder.org/) with the capacity to execute a model with the computational environment requirements analogous to the concept of analysis-ready data and cloud-optimized formats for datasets (Abernathey et al., 2021)."*

- [178] **Public availability of an (automated) testing suite**. I really like this point. I this is a good idea to look for automation of the tests. However, I would like to point out here that the concept of "automation" is in principle independent and complementary to the existence of the testing suite. Furthermore, the concept of automation applies also to some other indicators. For example, one can automate the creation of documentation using GitHub Pages, using GitHub actions to ensure the containerization of the software (even create all the ingredients for a docker file), and even ensure the compliance with coding standards of the software (see for example the Julia code formatter action: https://github.com/julia-actions/julia-format and I imagine there must be a way to automate the use of Pylint with Python).

Thank you for your feedback. We agree that the concept of automation is both independent and complementary to the existence of a testing suite. Since this section only accesses the availability of an (automated) testing suite, we have expanded our recommendations to include automation not only for testing but also for other aspects such as documentation creation, containerization, and coding standards compliance.

(Section 5, Line 567-571) *"Integrate automation in development practices. Automation plays a key role in streamlining software development by reducing manual effort and ensuring consistency (Wijendra and Hewagamage, 2021). We encourage developers to integrate automation into their workflows to improve efficiency. For instance, developers can use GitHub Actions to automate various tasks like running test suites, generating documentation, ensuring adherence to coding standards, and managing dependencies."*

- [269] How well documented are the models that just have a README file? In my experience, README documentations tend to be very plain and difficult to navigate. I think it is important to mention something about the quality of the documentation, at least as an observation.

Thank you for the comment. We have incorporated your suggestions and the full text now reads:

(Section 3, Line 288-294) *"Our analysis reveals that 75% of the GIMs (24 out of 32) have publicly accessible documentation (Table 2). We observed a range of documentation formats across these GIMs. Specifically, 6 GIMs provided readme files, 13 had dedicated webpages for documentation, and 5 included comprehensive manuals (see supplementary file ISIMIP_models.xlsx). While README files tend to be more minimal and sometimes difficult to navigate, we observed that they generally contain essential information such as instructions on how to run the research software. The prevalence of documentation practices among most models underscores the importance of documenting research software. However, a notable portion (25%) of the studied models either lack documentation or documentation has not been made publicly available (Table 2). "*

- [295] Following my previous comment, I think it is important to state why licensing is important and the difference between licenses. References or further support here is needed. Copyleft and permissive licenses are very different. I would also suggest pointing which one of the licenses in Figure 2 are copyleft of permissive.

Thank you for your valuable comment. We refer you to our response to [154] regarding the discussion of open-source licenses, where we expanded on the importance of licensing and the key differences between copyleft and permissive licenses, including references for further support. As the focus of our study is on the general use of open-source licenses rather than the specific restrictions associated with each type, we believe that distinguishing between copyleft and permissive licenses in Figure 2 would be outside the scope of this analysis.

- [378] I don't understand the purpose of this section. The effort here is calculated using equation 1, which (besides being based in many assumptions that the authors do take care of) already suggested that more lines of code mean more effort, which is an expected concussion. With this point in mind, I don't fully understand what new conclusion are made in Section 3.3. There is a chance that here I am missing an important point, and in that case, I would like the authors to clarify. I think here it would be interesting to see how the effort correlated with the number of satisfied indicators. Without further analysis, I would suggest removing or perform a different analysis in this section.

Thank you for your insightful feedback however the primary goal of Section 3.3 is not to draw new conclusions about the relationship between lines of code and effort, but rather to provide a rough effort estimate involved in developing these complex research software tools. This estimate aims to give developers and funders a sense of the scale of effort required, encouraging developers to invest in best practices for code development once funded, and making funders aware of the necessary support.

We have removed the first line of the paragraph, which we believe is confusing. We hope this clarification addresses your concerns. The revised now reads:

(Section 3, Line 404-419) *"To provide a rough cost estimate for the software development of the 32 impact models, we use the cost estimate model from Sachan et al. (2016) (see section 2.4) in a scenario of "what if we would hire a commercial software company to develop the source code of the global impact models?" This cost estimate does not include developing the science (e.g., concepts, algorithms, and input data) nor costs of documenting, running, and maintaining the software, only the implementation of code. We assume that the COCOMO model is transferable to research software as the NASA projects used in cost model contain software that is similar to research software. As the TLOC of the impact model codes ranges from 262 to 500,000 TLOC (Fig. 7), the effort required to produce these models ranges from 1 to 495 person-months (Fig. 7). With a small additive change of ±0.1 of the COCOMO model coefficients, the range of estimated effort changes to 1 to 255 person-months in the case of -0.1 , and to 1 to 960 person-months in the case of +0.1 (Supplementary Fig. S2).*

*The results suggest that these complex research software programs are expensive tools that require adequate funding for development and maintenance to make them sustainable. This is consistent with previous studies that have highlighted funding challenges for developing and maintaining sustainable research software in various domains (Carver et al., 2013, 2022; Eeuwijk et al., 2021; Merow et al., 2023; Reinecke et al., 2022). Merow et al. (2023) also emphasized that the accuracy and reproducibility of scientific results increasingly depend on updating and maintaining software. However, the incentive structure in academia for software development — and especially maintenance — is insufficient (Merow et al., 2023)."*

- [410] I think this is a very important question and should be first raised in lines 197-200, and maybe postponing the discussion until later. It is not clear for me that 30-60% is the desired number, since most of the time other heuristics are used for determining the number of comments. In a nutshell, better code requires minimal commenting and clarity in the code. (See book "Beautiful Code" for a nice collection of essays around this precise point.) Furthermore, I think is important to mention the entanglement between writing good documentation and commenting, since many software documentations are generated automatically based on the comments in the code (e.g., based on the docstrings in Python and Julia).

Thank you for your valuable feedback. Regarding minimal commenting on high-quality code, we agree that well-written code often requires fewer comments. However, it's important to note that the need for comments can vary depending on factors such as the programming language used, the complexity of the algorithms, and the expertise of the developers. Our paper specifically focuses on novice developers, particularly PhD students and postdocs in academic settings, who may not be expert programmers. In these environments, frequent turnover of personnel can result in new researchers inheriting poorly documented code, which can pose significant challenges. Therefore, while minimal commenting may be appropriate for highly experienced developers, the context of academic research and novice coders often requires more explicit comments for clarity and maintainability.

Regarding the comment density recommendation, we acknowledge that our initial phrasing around the 30-60% comment density may have come across as too prescriptive. Our intention was to reference this range as it is commonly cited in the literature, not to imply it as a strict rule. We have revised our text to clarify this, now stating:

(Section 2, Line 216-219) " . *Arafat et al. (2009) and He (2019) suggest that comment density between 30-60% may be optimal. For most programming languages, this range is considered to represent a compromise between providing sufficient comments for code explanation and having too many comments that may distract from the code logic (Arafat and Riehle, 2009; He, 2019).* "

We recognize that not all programming languages natively support automatic generation of documentation from comments. Our discussion of this feature now intends to highlight its potential benefits where available, rather than suggest it as a universal solution (see Section 5, line 567 on automation).

- [490] Here it is mentioned a very important point: design principles, or design patterns. More than writing code that follows certain standards, it is important to think about the overall software architecture of the model. E.g., if I am working in Python, what are going to be my classes? How do they interact with each other? How data will be processed? Etc. None of the indicators really address this aspect of software development (which I think is fine for the scope of the manuscript), but I think the authors should emphasize this point in the recommendation section.

Thank you for your comment. We now emphasise the points suggested. The section now reads

(Section 5, Line 541-544) *"Design principles help adhere to the principles of sustainable research software, such as modularity, reusability and interoperability. These principles also*

*guide the design of software by determining, for instance, the interaction of classes addressing aspects such as separation of concerns, abstraction, and encapsulation (Plösch et al., 2016)."*

- [496] As it was already mentioned, I think is important to remark that what here is referred as internal and external documentation sometimes are the same. Documentation can be created from comments in the code, specially docstrings, and this is actually a great documentation practice, to make the internal and external documentation to be the same so there is no repetition nor contradiction between the two.

Thank you for your feedback. While we recognize external documentation can be generated from code comments, we respectfully disagree that internal and external documentation should always be the same. Not all programming languages support this feature, and external documentation often includes additional resources such as videos, publications, tutorials (as discussed in section 5, lines 550) that go beyond what is covered in docstrings.

- Since automation is mentioned as playing an important role, I would suggest including automation in a more general sense in the recommendation section. This includes using GitHub Actions to run the test suite, automate the creation of the documentation (static or dynamic), check for code style, check packages dependencies, etc. Another interesting tool the authors may want to consider mentioning is Makefile.
We refer you to our response to [178] regarding the discussion of automation.

- Since the paper addresses the important aspect or sustainable software, I would strongly suggest that the code used to generate the analysis, and the figures of the manuscript are presented in the same standard that the nine indicators dictate. I think this will really improve the quality of the work, and it can serve as an example in the manuscript itself of "how things should be done". I think readers would like to see that, and I think it is part of the philosophy of the manuscript to promote these good practices. Furthermore, I would make the point that the same tools that had been used in this manuscript can be used for analyzing other source code models, so the code used in the manuscript can be re-usable by other users.
Thank you for the suggestion. While we understand the desire to showcase our analysis code as an example, the scripts (although developed with the best practices) used for data processing and visualization differ significantly from the complex research software models the nine indicators are designed for. These simpler scripts don't require the same architectural planning or extensive documentation, nor do they fully embody indicators like testing or licensing. However, for those interested in sustainable research software, models like HydroPy (see section 3.4) are a great starting point. This can serve as a better example of how to apply the indicators discussed in the manuscript.
We now state that in the revised manuscript (Section 3, Line 445-446)

*"The HydroPy model is great starting point for sustainable research software development as it illustrate the application of the sustainability indicators."*

**Minor Comments**

- [30] I am a bit confused by what Earth system modelling entitles. When presenting the models, we are talking about models in agriculture, biomes, fire, etc. In line 33, it says that "While so-called Earth System Models always include the simulation of atmospheric processes and thus

compute climate variables and how they change due to greenhouse gas emissions […]". However, this excludes many "Earth system models", including all those not based in atmospheric processes. I think the phrasing of Earth system model should be narrowed to what the models are for. For example, all the models in the ISIMIP are about the impact of climate change.

Thank you for your comment. We have now revised the line 33 for clarity which now reads:

(Section 1, Line 33-35) *"A specific class of simulation models of the Earth called impact models enables us to quantitatively estimate the potential impacts of climate change on, e.g., floods (Sauer et al., 2021), droughts (Satoh et al., 2022), and food security (Schmidhuber and Tubiello, 2007)."*

- [66-70] Repeats reference Anzt et. al. (2021)

We have revised this section for clarity, referring to the definition by the authors: Anzt et al. (2021), which now reads:

(Section 1, Line 68-71) *"Anzt et al. (2021) define research software as software that is maintainable, extensible, flexible (adapts to user requirements), has a defined software architecture, is testable, has comprehensive in-code and external documentation, and is accessible (the software is licensed as Open Source with a digital object identifier (DOI) for proper attribution) (Anzt et al., 2021)."*

- [83-99] I suggest splitting this paragraph into two, with the second stating what is done specifically in this work (maybe just break before "In this study, we assess…").

We have split the paragraph into two as suggested. The paragraph with "In this study, we assess…" snow starts from Section 1 Lines 96.

- [105] Data in this line means "model"? I will suggest not to use "data" to refer to the models in the ISIMIP database, since it is a bit confusing. If this is referring to another type of data, maybe explain.

Thank you for the comment. We have revised the section accordingly:

(Section 2, Line 111-112) "*As the focus of our analysis is on global impact models, we sorted the models by spatial domain and filtered out models operating at local and regional scales, resulting in a subset of 264 GIMs.*"

- [110-111] This sentence requires rephasing, since it is ambiguous what it means by "in the described way" (meaning GitHub/GitLab or also including source code in reference papers).

We have rephrased this section to remove ambiguity. This section now reads.

(Section 2, Line 116-118) *"As of April 2024, 32 out of the 112 unique model source codes were accessible either through direct links from the ISIMIP database or via manual searches on platforms like GitHub and GitLab, as well as in code availability sections of reference papers."*

- [189] Is PEP8 the de-facto coding style in Python? I think there used to be some alternatives and may be important to mention something like this. Furthermore, in other programming languages there are more than one coding style that are accepted by the community (e.g., in

Julia), so it may be important to mention that the important aspect of this is that the developers of one model stick with one style, rather that sticking with one single style.

PEP8 is widely regarded as the standard style guide for Python (https://arxiv.org/pdf/2408.14566), although some organizations, such as Google, have their own internal versions. In Line 189, which focuses solely on the methods, we only discuss the process of accessing coding standards and, for simplicity, concentrate on PEP8 in Python, as there are tools for measuring compliance to PEP8 standard. As suggested, we now note that there can be more than one coding style that is accepted by the language community (e.g., Julia). The revised section now reads:

(Section 2, Line 202-204) *"Analysing the conformance to these standards can be complex, particularly when the source code is written in multiple languages. Different languages may have various coding styles or style guides. For instance, multiple style guides are available and accepted by the Julia community (JuliaReachDevDocs, 2024)."*

We agree that developers consistently follow one coding style for their project hence we revise or recommendation section to clearly state this.

(Section 5, Line 537-540) *"Use coding standards accepted by your community (e.g., PEP8 for Python), good and consistent variable names, design principles, code quality metrics, peer code review, linters and software testing: Coding standards help you write clear, consistent, and readable code that follows the best practices of your programming language and domain. It is key that developers consistently follow a coding style recognized by the relevant language community. …"*

- Table 1. Following the logic in the text, I will suggest making the division in the table between best practice in software engineering and source code quality (e.g., adding this information in the table, dividing them with a horizontal line). I think this will make the conceptual difference clearer than using the footnote. Check punctuation at the end of the descriptions. Some of the items end with dot, not dot, or comma.
Thank you very much for the feedback. Table 1 has been revised as suggested.

- [245] These are the 32 models mentioned in line 110, right? I would mention this again for clarity.
Thank you for your comment. We have already clarified this in the relevant section. Specifically, we mention:
(Section 3, Lines 263-264) *"The source code of the 32 GIMs is written in 10 programming languages (Fig. 1a). Fortran and Python are the most widely used, with 11 and 10 models, respectively."*

- Figure 1. I will suggest sorting the bars in increasing/decreasing order. This is a comment I had with all the rest of the tables of the paper, where I would order the bars for clarity.
We have revised Figures 1, 2, 3, 4, 5, 7 to sort the bars in decreasing order for clarity. Note that for Figures 3, 5 and 7, we have sorted the data by decreasing values within each sector.

- [255] I will suggest not starting the sentence with a numeral since this is uncommon and non recommended in formal English.

We have revised the sentence accordingly.

(Section 3, Line 274) *"We find that 24 (75%) of the readily accessible 32 GIMs were hosted on GitHub (Fig. 1b)."*

- [289] There is some repetition between what is said here and the paragraph in line 255 and Figure 1. I think mentioning git and the corresponding platform (GitHub, …) should be made once in the same section for clarity in the text.
  We have revised the section accordingly.  This now reads:

  (Section 3, Line 309-311) *"We find that 81% (26 out of 32) of GIMs uses Git as their version control system reflecting the widespread acceptance of Git across the sectors (Table 2).  In the remaining cases, information about the specific version control system used for these GIMs was unavailable."*

- Consider introducing Table 2 before Figure 1, since it contains the overall information of the model.
  Thank you for your suggestion, however, we prefer to keep the current order, with Figure 1 introduced before Table 2, as it aligns better with the overall flow and structure of the paper.

- It would be great to add an extra column to Table 2 with the year of the model (last version) and sort by this. I think it would be interesting to see if the availability of documentation, version control, etc, had improved over the years. Also interesting to see programming language used and how this changed over the years.
  Thank you for your valuable suggestion. We have added an extra column to Table 2 to include the year of the model's last version. However, we found that sorting the models by year resulted in poor readability. Therefore, we have maintained the sorting by sectors, as it aligns better with the logic of most figures.

- [291] I don't agree with the statement that "Developers' preference for Git highlights its user-friendly nature and effectiveness in supporting collaborative efforts". I think there may be other reasons for this, since there were and are other version control systems that are as user-friendly as git but didn't became that popular. One reason for this is that GitHub naturally uses git, and that many developers use VSCode with also supports git and GitHub. If the authors want to keep this sentence as it is, the statement should be supported by references.
  Thank you for your feedback. We have considered your comments and have decided to remove the statement regarding "developers' preference for Git…". (Section 3, Line 308)

- [306] It is important to start saying what is a good number of contributors, or what is expected to see here. It is unclear to me that ~10 contributors is robust enough.
  Thank you for your comment. We do not specify a required number of developers for a project to be considered robust, as this can vary significantly depending on the project's size and complexity. While a larger number of active developers may indicate a robust and well-maintained project, it is not a strict requirement for all GIMs. Smaller or less complex projects can be effectively maintained by even a single experienced developer.

Our measure of active developers serves as an indicator of ongoing maintenance and the prevention of software stagnation, rather than a definitive measure of a project's robustness. We have stated the goal of using this indicator in the method section (Section 2, Line 174)

- [320] Mention which container platform these 5 models used. Did all used Docker? If so, how do they share it?
Thank you for the suggestion. Apart from the CLASSIC model, which uses Apptainer, the remaining four models utilize Docker as their containerization technology. The CLASSIC container is shared via Zenodo, whereas the Docker containers for the other four models are distributed through GitHub. We have updated our text to reflect these details:

(Section 3, Line 340-344) *"Only 5 (16%) of the GIMs have implemented containerized solutions (Table 2). While the CLASSIC model uses Apptainer, the other four models use Docker as their containerization technology. The CLASSIC container is shared via Zenodo, whereas the Docker containers for the remaining models are distributed through GitHub. Despite the recognized benefits of containerization in promoting reproducible research, provisioning of the software in containers is not yet a common practice in GIM development."*

- [326] I am curious: do the models with test suite use a preferred programming language? Does the programming language plays a role in how easy is to implement the test suite? Maybe the authors want to answer to this question in the manuscript, I think it will make an interesting point.
Thank you for the suggestion. We now answer both questions in the manuscript.

(Section 3, Line 349-354) *"Our research indicates that 28% (9 out of 32) of the examined GIMs have a testing suite in place to test the software's functionality (Table 2). The models with test suites do not use a preferred programming language but have various languages, including Python, Fortran, R, and C++ (Table 2). While the choice of programming language can influence the ease of implementing test suites (e.g., due to the availability of testing libraries), we observe that for these complex models, which often prioritize computational performance, implementing a test suite remains essential regardless of the programming language used."*

- [249] I will suggest not starting the sentence with a numeral.
Thank you for your comment. The actual line is 349. We have revised the sentence accordingly and all other occurrences in the paper.

(Section 3, Line 375-376) *"Our results indicate that 25% (8 of 32) of the GIMs have well-commented source code, i.e. 30-60% of all source lines of code are comment lines (Fig. 5)."*

- [429] I really enjoyed this section, and I think it will improve to the communication of the paper to put this section earlier in the manuscript, maybe between the introduction of the indicators and the analysis and the results.
Thank you for your positive feedback on this section. We appreciate your suggestion to move it earlier in the manuscript. However, we believe that its current placement aligns with the overall logic and structure of the paper.

- [478] Same point about permissive and copyleft licenses. What do the authors mean by open-source licenses? Do you mean permissive?

  Thank you for your valuable comment. We refer you to our response to [154] regarding the discussion of open-source licenses

- [482] I am not sure that version control ensures software reproducibility, not with other important tools (environment or containerization, testing suite, etc).

  Thank you for your feedback. We agree with the reviewer that version control alone does not guarantee software reproducibility. As noted by Jiménez et al. (2017), version control facilitates the reproducibility of scientific results generated by all prior versions of the software. Therefore, we have revised our statement as follows:

  *(Section 5, Line 531-533) "Version control can help you track and manage changes to your source code, which ensures the traceability of your software and facilitates reproducibility of scientific results generated by all prior versions of the software (Jiménez et al., 2017)."*

General comments

- I strongly suggest sorting all the tables in decreasing or increasing order for readability.

  Thank you for the comment. All relevant tables including figures have been sorted accordingly.

- I suggest the authors to do a English style revision of the manuscript since there are different styles in the manuscript. This includes the starting of the sentences with numerals and the use or not use of contractions.

  We have revised relevant sentences that occur in the paper.

- In the same style that Figure 9, I think a figure at the beginning of the manuscript summarizing the indicators and what are good software practices will improve the manuscript.

  Thank you for the suggestion, however, we believe that Table 1 already effectively summarizes the indicators and good software practices.

[revised manuscript text omitted]

---

## Author Comment (AC3)

**Reply to Community Comment: Tijn Berends**

Dear Reviewer,

We appreciate your prompt and critical review of our paper. Your thoughtful comments and suggestions have greatly improved the quality of our revised manuscript.

In the following sections, we have addressed your comments point-by-point and changed the manuscript accordingly. Please find the tracked changes attached to this letter. Your suggestions are highlighted in blue, while our responses are in black and new text italics. All section and line numbers mentioned correspond to the revised manuscript. In summary, we have revised aspects of the introduction and enhanced the results and discussion sections.

I am glad to see a manuscript like this. With the ever-increasing political and societal demand for new, more accurate scientific knowledge about the Earth system, and particularly its future state, the complexity of computational models has grown exponentially over the past few decades. The need for software engineering skills, on top of the knowledge of the scientific domain, and the wide and varying set of skills required of an active research scientist, is now an undeniable reality. New literature investigating just exactly what "software engineering skills" entails in the context of research software is therefore a valuable addition.

Thank you for responding to our manuscript and highlighting its timeliness. This really shows the value of the community comment functionality.

Having the luxury of being an uninvited reviewer, I can constrain myself to only pointing out the bits that really strike me, and leave the detailed feedback to the invited reviewers. Two points stand out to me in this manuscript that I think could do with some improvement.

Firstly, there is the concept of "self-explanatory code" mentioned in lines 406-413. While I appreciate that the authors are merely citing another group's description of that group's own work, I believe this statement needs a disclaimer. Depending on how you define "self-explanatory", either all code qualifies as such, or none. If, for example, we define code as "self-explanatory" when one can *eventually* arrive at an understanding of its functionality without consulting the original author, then all code is self-explanatory – with, of course, the caveat that "eventually" can, in many cases, be prohibitively far into the future. On the other hand, if we define code as "self-explanatory" when we require no other resources to (again, eventually) understand its functionality, then probably no code ever meets this definition, at least in the context of research software, which always requires a substantial level of background knowledge on the part of the developer. E.g., is the code that calculates the sea-level equivalent volume of an ice sheet truly self-explanatory if it does not explain the concept of sea-level equivalent volume? In my view, these considerations illustrate that the phrase "self-explanatory code" is so difficult to define as to be practically meaningless. In my experience, it is used mainly by people who inherited code from their supervisor that is not as well-commented as they'd like it to be, but cannot say so out loud for fear of their career prospects. I'm sure the authors can add these considerations, possibly in a rephrased manner, to their revised manuscript.

We acknowledge that the term "self-explanatory code" can indeed be interpreted in various ways, which may render it problematic without further clarification. We have revised the manuscript to address the complexities surrounding code readability and the need for comments. The updated text now reads.

(Section 3, Line 436-444) *"MPI-HM has more comments (49%, Fig. 8b) because of its legacy Fortran code that limits variable names to a maximum length of 8 characters, so they have to be described in comments. Another reason is that the MPI-HM developers kept track of the file history in the header, which adds to the comment lines in MPI-HM. This raises a question: Is the comment density threshold metric still valid if a code is highly readable and comprehensive? The need for comments can depend on the language's readability (Python vs. Fortran), the complexity of the implemented algorithms and concepts, and the coder's expertise. While a highly readable and well-structured code might require fewer explanatory comments, the definition of "readable" itself can be subjective and context-dependent. Nevertheless, comment density remains a valuable metric, especially for code written by novice developers."*

Secondly, there is the first of the authors' recommended best practices, in lines 470-473, where they support the use of Agile as a project management framework. Having briefly worked at a company that applied this framework (and much longer as a researcher building my own numerical models), I have some small amount of experience with it, and I must say I do not immediately see its value in a research setting. The highly individualistic nature of scientific research(ers), the very poorly-defined goals, scope, and expected duration of research projects, as well as the high degree of overlap between developers, managers, stakeholders, and users, make the use of Agile difficult in a research context. Additionally, while I see how Agile working can improve the speed with which a large group of people produce output within a certain project, I do not immediately see how Agile affects the *quality* of that output – which is the subject of this manuscript. I.e., agile scientists might produce science faster, but not necessarily better. If the authors can provide arguments to the contrary, I'd happily read them, but right now nothing of the sort is written in the manuscript – in fact, the concept of Agile is only mentioned briefly at one point in the introduction, and then does not appear again until it is listed as the first "recommended best practice". I hope the authors can remedy this lack of evidence for their claim of Agile being a "best practice" in the revised version of their manuscript.

Thank you for the valuable feedback. We agree with the reviewer, particularly regarding the limitations of Agile in certain scientific settings, such as those with fixed requirements. However, we believe that some principles and practices from Agile can still provide significant value in academic software development, particularly in environments where timelines are constrained and requirements do change quickly (with newly available data or methods), such as PhD and Postdoc positions. Below, we outline some key benefits:

- Flexibility in Handling Evolving Requirements: While many scientific projects have well-defined requirements, others benefit from Agile's adaptability to changing conditions. For example, as new research questions arise or additional factors need to be considered (e.g., integrating karst regions into hydrological research software for groundwater recharge), Agile's iterative process allows for the efficient incorporation of these updates. This flexibility ensures that the software remains responsive to ongoing developments.
- Transparency and Progress Tracking: Agile's use of tools like backlogs, task boards, and regular progress updates provides clear visibility into the project's status. This is particularly useful in academic settings, where project continuity can be disrupted due to personnel changes, such as the arrival or departure of researchers. These tools help ensure smooth transitions by clearly indicating completed tasks and those still in progress, thereby minimizing disruptions to the workflow.
- Enhanced Collaboration and Communication: Agile emphasizes frequent communication among team members and stakeholders, which can be highly beneficial in academic settings.

Regular meetings and updates help ensure alignment among diverse contributors, such as students, researchers, and supervisors. This ongoing collaboration helps keep the team informed and engaged, and allows for timely input and feedback as the project evolves.

To be more nuanced about our recommendation we have revised our recommendation on project management practices to

[revised manuscript text omitted]

---

## Author Comment (AC4)

**Reply to Community Comment: Tijn Berends**

Dear Reviewer,

We appreciate your prompt and critical review of our paper. Your thoughtful comments and suggestions have greatly improved the quality of our revised manuscript.

In the following sections, we have addressed your comments point-by-point and changed the manuscript accordingly. Please find the tracked changes attached to this letter. Your suggestions are highlighted in blue, while our responses are in black and new text italics. All section and line numbers mentioned correspond to the revised manuscript. In summary, we have revised aspects of the introduction and enhanced the results and discussion sections.

I am glad to see a manuscript like this. With the ever-increasing political and societal demand for new, more accurate scientific knowledge about the Earth system, and particularly its future state, the complexity of computational models has grown exponentially over the past few decades. The need for software engineering skills, on top of the knowledge of the scientific domain, and the wide and varying set of skills required of an active research scientist, is now an undeniable reality. New literature investigating just exactly what "software engineering skills" entails in the context of research software is therefore a valuable addition.

Thank you for responding to our manuscript and highlighting its timeliness. This really shows the value of the community comment functionality.

Having the luxury of being an uninvited reviewer, I can constrain myself to only pointing out the bits that really strike me, and leave the detailed feedback to the invited reviewers. Two points stand out to me in this manuscript that I think could do with some improvement.

Firstly, there is the concept of "self-explanatory code" mentioned in lines 406-413. While I appreciate that the authors are merely citing another group's description of that group's own work, I believe this statement needs a disclaimer. Depending on how you define "self-explanatory", either all code qualifies as such, or none. If, for example, we define code as "self-explanatory" when one can eventually arrive at an understanding of its functionality without consulting the original author, then all code is self-explanatory – with, of course, the caveat that "eventually" can, in many cases, be prohibitively far into the future. On the other hand, if we define code as "self-explanatory" when we require no other resources to (again, eventually) understand its functionality, then probably no code ever meets this definition, at least in the context of research software, which always requires a substantial level of background knowledge on the part of the developer. E.g., is the code that calculates the sea-level equivalent volume of an ice sheet truly self-explanatory if it does not explain the concept of sea-level equivalent volume? In my view, these considerations illustrate that the phrase "self-explanatory code" is so difficult to define as to be practically meaningless. In my experience, it is used mainly by people who inherited code from their supervisor that is not as well-commented as they'd like it to be, but cannot say so out loud for fear of their career prospects. I'm sure the authors can add these considerations, possibly in a rephrased manner, to their revised manuscript.

We acknowledge that the term "self-explanatory code" can indeed be interpreted in various ways, which may render it problematic without further clarification. We have revised the manuscript to address the complexities surrounding code readability and the need for comments. The updated text now reads.

(Section 3, Line 436-444) "MPI-HM has more comments (49%, Fig. 8b) because of its legacy Fortran code that limits variable names to a maximum length of 8 characters, so they have to be described in comments. Another reason is that the MPI-HM developers kept track of the file history in the header, which adds to the comment lines in MPI-HM. This raises a question: Is the comment density threshold metric still valid if a code is highly readable and comprehensive? The need for comments can depend on the language's readability (Python vs. Fortran), the complexity of the implemented algorithms and concepts, and the coder's expertise. While a highly readable and well-structured code might require fewer explanatory comments, the definition of "readable" itself can be subjective and context-dependent. Nevertheless, comment density remains a valuable metric, especially for code written by novice developers."

Secondly, there is the first of the authors' recommended best practices, in lines 470-473, where they support the use of Agile as a project management framework. Having briefly worked at a company that applied this framework (and much longer as a researcher building my own numerical models), I have some small amount of experience with it, and I must say I do not immediately see its value in a research setting. The highly individualistic nature of scientific research(ers), the very poorly-defined goals, scope, and expected duration of research projects, as well as the high degree of overlap between developers, managers, stakeholders, and users, make the use of Agile difficult in a research context. Additionally, while I see how Agile working can improve the speed with which a large group of people produce output within a certain project, I do not immediately see how Agile affects the *quality* of that output – which is the subject of this manuscript. I.e., agile scientists might produce science faster, but not necessarily better. If the authors can provide arguments to the contrary, I'd happily read them, but right now nothing of the sort is written in the manuscript – in fact, the concept of Agile is only mentioned briefly at one point in the introduction, and then does not appear again until it is listed as the first "recommended best practice" in the revised version of their manuscript.

Thank you for the valuable feedback. We agree with the reviewer, particularly regarding the limitations of Agile in certain scientific settings, such as those with fixed requirements. However, we believe that some principles and practices from Agile can still provide significant value in academic software development, particularly in environments where timelines are constrained and requirements do change quickly (with newly available data or methods), such as PhD and Postdoc positions. Below, we outline some key benefits:

- Flexibility in Handling Evolving Requirements: While many scientific projects have welldefined requirements, others benefit from Agile's adaptability to changing conditions. For example, as new research questions arise or additional factors need to be considered (e.g., integrating karst regions into hydrological research software for groundwater recharge), Agile's iterative process allows for the efficient incorporation of these updates. This flexibility ensures that the software remains responsive to ongoing developments.
- Transparency and Progress Tracking: Agile's use of tools like backlogs, task boards, and regular progress updates provides clear visibility into the project's status. This is particularly useful in academic settings, where project continuity can be disrupted due to personnel changes, such as the arrival or departure of researchers. These tools help ensure smooth transitions by clearly indicating completed tasks and those still in progress, thereby minimizing disruptions to the workflow.
- Enhanced Collaboration and Communication: Agile emphasizes frequent communication among team members and stakeholders, which can be highly beneficial in academic settings.

Regular meetings and updates help ensure alignment among diverse contributors, such as students, researchers, and supervisors. This ongoing collaboration helps keep the team informed and engaged, and allows for timely input and feedback as the project evolves.

To be more nuanced about our recommendation we have revised our recommendation on project management practices to

(Section 5, Line 515-522) "Choose project management practices that align with your institutional environment, culture, and project requirements: This can help plan, organize, and monitor your software development process, as well as improve collaboration and communication within your team and with stakeholders. Project management practices also help identify and mitigate risks, manage changes, and deliver quality software on time and within budget (Anzt et al., 2021). While traditional methods may be better suited for projects with fixed requirements, certain principles from more flexible frameworks, such as Agile, can provide benefits in environments where requirements evolve or adaptability is critical. For example, Agile's iterative approach allows for incorporating changing research questions and hence software modifications or extensions, improving responsiveness to new developments (Turk et al., 2005)."

**Software sustainability of global impact models**

Emmanuel Nyenah1, Petra Döll1, 2, Daniel S. Katz3, and Robert Reinecke4

[revised manuscript text omitted]

| No. | Sector               | Model                   | Year of           | Language                 | Documentation | Version        | Open           | Test           | Container | Formatted: Font: 8 pt                           |
|-----|----------------------|-------------------------|-------------------|--------------------------|---------------|----------------|----------------|----------------|-----------|-------------------------------------------------|
|     |                      |                         | Latest
Version |                          |               | control        | Source         | Suite          |           | Eormatted Table                                 |
| 1   | Agriculture          | CGMS-WOFOST             | no info           | Fortran           | <del>*±</del> | *+      | ×±             | -              | -         | Formatted: Fast: 9 st. English (United Kingdom) |
| 2   | Agriculture          | DSSAT-Pythia            | 2024              | Python                   | <del>x+</del> | <del>x</del> + | no info        | <del>x.+</del> | *+        |                                                 |
| 3   | Agriculture          | EPIC-TAMU               | 2023              | Fortran                  | *+            | no info        | *+             | -              | _         | Formatted: Font: 8 pt                           |
| 4   | Agriculture          | I PImI                  | 2024              | Cand                     | ar. 1.        | ar L           | ar. 1.         | _              |           | Formatted: Font: 8 pt                           |
| -   | Agriculture          | Ersnie                  | 2024              | JavaScript               | AT            |         |         | -              |           | Formatted: Font: 8 pt                           |
| 5   | Agriculture          | ACEA                    | 2024              | Python                   | * +    | no info        | not valid      | -              | `         | Formatted: Font: 8 pt                           |
| 6   | Agriculture          | LPJ-GUESS               | 2021       | C++               | *±     | no info        | *±      | -              | -         | Formatted: Font: 8 pt                           |
| 7   | Biomes               | CLASSIC                 | 2020              | Fortran                  | *+     | *+      | <del>*+</del>  | *+      | *+ | Formatted: Font: 8 pt                           |
| 8   | Biomes               | MC2-USFS-r87g5c1        | 2022              | C++,              | * +    | *+      | *+      | -              | -         | Formatted: Font: 8 pt                           |
|     |                      |                         |                   | and C                    |               |                |                |                |           | Formatted: Font: 8 pt                           |
| 9   | Fire                 | SSiB4/TRIFFID-Fire      | 2021              | Fortran                  | -             | *+      | no info        | -              | -         | Formatted: Font: 8 pt                           |
| 10  | Fisheries            | BOATS                   | no info    | MATLAB                   | -             | *+      | no info        | -              | -         | Formatted: Font: 8 pt                           |
| 11  | Fisheries            | DBPM                    | no info    | R                 | -             | *+      | no info        | *+      | -         | Formatted: Font: 8 pt                           |
| 12  | Fisheries            | EcoTroph                | no info    | R                 | *+     | *+      | no info        | -              | -         | Formatted: Font: 8 pt                           |
| 13  | Fisheries            | FEISTY                  | no info           | MATLAB                   | -             | *+      | no info        | -              | -         | Formatted: Font: 8 pt                           |
| 14  | Fisheries            | ZooMSS                  | 2020              | R and c++                | * +    | *+      | <del>*+</del>  | -              | -         | Formatted: Font: 8 pt                           |
| 15  | Groundwater          | G 3 M        | 2018              | C++               | *+     | *+      | <del>x_+</del> | *+      | -         | Formatted: Font: 8 pt                           |
| 16  | Groundwater          | parflow                 | 2024              | C, Tcl,           | *+     | *+      | *+      | *+      | *± | Formatted: Font: 8 pt                           |
| 17  | Lakes                | ALBM                    | 2024              | python
Fortran | <del>x+</del> | <del>x</del> + | <del>x+</del>  | -              | _         | Formatted: Font: 8 pt                           |
| 18  | Lakes                | GOTM                    | 2024              | Fortran                  | *+            | <del>x+</del>  | *+             | <del>x+</del>  | -         | Formatted: Font: 8 pt                           |
| 19  | Lakes                | SIMSTRAT-UoG            | 2024              | Fortran                  | *+            | <del>x+</del>  | *+             | <del>x.+</del> | *+        | Formatted: Font: 8 pt                           |
| 20  | Terrestrial          | BioScen15-SDM-GAM/GBM   | no info           | R                        | -             | *+             | no info        | -              | _         | Formatted: Font: 8 pt                           |
|     | biodiversity         |                         |                   |                          |               | _              |                |                |           | Formattedi Fonti 9 pt                           |
| 21  | biodiversitv         | BIOSCEN 1.5-MEM-GAM/GBM | no into    | K                        | -             | *+      | *+      | -              | -         |                                                 |
| 22  | Vector-borne         | VECTRI                  | no info    | Fortran                  | *+     | *+      | <del>x_+</del> | -              | -         | Formatted: Font: 8 pt                           |
|     | diseases
(health) |                         |                   | and
python            |               |                |                |                |           | Formatted: Font: 8 pt                           |
| 23  | Water                | CWatM                   | 2023              | Python                   | *+     | *+      | <del>*±</del>  | *+      | -         | Formatted: Font: 8 pt                           |

 Table 2: Availability of Documentation, Version Control, Open-Source License, Test Suite, and Container for 32 Global

 [315] Impact Models across 10 Sectors in Earth System Science. '+\*, '-', 'not valid' and 'no info' represent the availability, unavailability, not OSI-approved and absence of information, respectively.

| 24 | Water         | DBH          | 2006           | Fortran           | *+  | no info        | not valid     | -         | -          |    | Formatted: Font: 8 pt |
|----|---------------|--------------|----------------|-------------------|------------|----------------|---------------|-----------|------------|----|-----------------------|
| 25 | Water         | HydroPy      | 2021    | Python            | *+  | no info        | *+     | -         | -          |    | Formatted: Font: 8 pt |
| 26 | Water         | PCR-GLOBWB   | 2023           | Python            | *+  | <del>*</del> + | <del>*+</del> | -         | -          | -1 | Formatted: Font: 8 pt |
| 27 | Water         | WBM          | 2023           | Perl              | *+  | *+      | *+     | -         | -          | -( | Formatted: Font: 8 pt |
| 28 | Water         | WaterGAP2.2e | 2023           | C++        | -          | no info        | *+     | -         | -          | -( | Formatted: Font: 8 pt |
| 29 | Water         | VIC          | 2021    | C and             | * + | *+      | *+     | *+ | * + |    | Formatted: Font: 8 pt |
| 30 | Water         | H08          | 2024           | Python
Fortran | *+  | *+      | *+     | -         | -          | (  | Formatted: Font: 8 pt |
| 31 | Water         | WAYS         | no info | Python            | -          | *+      | *+     | -         | -          | (  | Formatted: Font: 8 pt |
| 32 | Water quality | DynQual      | 2023           | Python            | *+  | *+      | no info       | -         | -          | -( | Formatted: Font: 8 pt |
| A  |               | Total        |                |                   | 24         | 26             | 22            | 9         | 5          | -  | Formatted: Font: 8 pt |

Version control:

 We find that 81% (26 out of 32) of GIMs uses Git as their version control system reflecting the widespread acceptance of Git
 across the sectors (Table 2). In the remaining cases, GIMs were made available on Zenodo and the models' official websites (Table 2, Fig. 1b); information about the specific version control system used for these GIMs was unavailable. Developers' preference for Git highlights its user-friendly nature and effectiveness in supporting collaborative efforts.

Use of an open source license:

325 Most of the research software, 69% (22 out of 32), have open-source licenses (Table 2) with the "GNU General Public License" being the commonly used license (56%, 18 out of 32) (Fig. 2). However, the remaining 31% (10 out of 32) either have no information on the license even though the source code is made publicly available (8 or 25% of GIMs) or uses license which is not OSI-approved (1 GIM each with creative commons license and user agreement) (Fig. 2). This ambiguity or absence of licensing details can deter potential users and contributors, as it raises uncertainties about the permissions and restrictions associated with the software.

Figure 2: License distribution for 32 global impact models across 10 sectors. 8 (25%) GIMs lack license information, and two (6%) GIMs have licenses that are not OSI-approved.

335

**Number of active developers:**

Our results reveal a diverse distribution of active developers across the GIMs. We have excluded GIMs without version control information from our results, as without could not be evaluated for this indicator, resulting in data for 26 GIMs. Notably, GIMs such as parflow, CWatM, LPJmL, and GOTM have a significant number of active developers, with 28, 12, 11, and 8 developers
 340 respectively (Fig. 3). These values correlates with the size of GIMs source code, as evidenced by TLOC (282,722 for ParFlow, 33,286 for CWatM, 136,002 for LPJmL, and 29,477 for GOTM.). However, models such as WAYS, VIC, BioScen1.5-MEM,

and CGMS-WOFOST had no active developers during the considered period of 2020 to 2024 (Fig. 3).